# Exploring Collaboration Mechanisms for LLM Agents: A Social Psychology View

## Abstract

As Natural Language Processing (NLP) systems are increasingly employed in intricate social environments, a pressing query emerges: *Can these NLP systems mirror human-esque collaborative intelligence, in a multi-agent society consisting of multiple large language models (LLMs)?* This paper probes the collaboration mechanisms among contemporary NLP systems by melding practical experiments with theoretical insights. We fabricate four unique 'societies' comprised of LLM agents, where each agent is characterized by a specific 'trait' (easy-going or overconfident) and engages in collaboration with a distinct 'thinking pattern' (debate or reflection). Evaluating these multi-agent societies on three benchmark datasets, we discern that LLM agents navigate tasks by leveraging diverse social behaviors, from active debates to introspective reflections. Notably, certain collaborative strategies only optimize efficiency (using *fewer* API tokens), but also outshine previous top-tier approaches. Moreover, our results further illustrate that LLM agents manifest human-like social behaviors, such as conformity or majority rule, mirroring foundational Social Psychology theories. In conclusion, we integrate insights from Social Psychology to contextualize the collaboration of LLM agents, inspiring further investigations into the collaboration mechanism for LLMs. We commit to sharing our code and datasets (already submitted in supplementary materials), hoping to catalyze further research in this promising avenue.

## 1 Introduction

With the prevalence of LLMs (Zhao et al., 2023; Yin et al., 2023; Zhu et al., 2023) integral to daily social collaboration, there's a growing imperative to cultivate AI systems embodied with social intelligence. This also resonates with the Society of Mind (SoM) concept (Zhuge et al., 2023; Wang et al., 2023c), which suggests that intelligence emerges when computational modules interact with each other, achieving collective objectives that surpass the capabilities of individual modules (Minsky, 1988). Previous studies (Park et al., 2023; Li et al., 2023; Du et al., 2023; Liang et al., 2023; Hao et al., 2023; Liu et al., 2023a; Akata et al., 2023) have delved into strategies where LLM instances, termed agents (Xi et al., 2023; Wang et al., 2023a), cooperate synergistically (e.g., debate and reflect) to accomplish tasks. As illustrated in Figure 1, such collaboration fosters divergent thinking processes in LLMs, making them particularly effective for tasks demanding profound reflection.

Intuitively, reflecting on human societies (Sap et al., 2022; Shapira et al., 2023), where a myriad of individual with distinct goals and roles coexist, the SoM framework champions harmonious interactions (Singh, 2003). Intriguingly, despite the fusion of social psychology (Tajfel, 1982; Tajfel & Turner, 2004) in SoM with human group dynamics (Woolley et al., 2010), which illuminates psychological patterns within social groups, its interpretation in the realm of LLMs is relatively uncharted. Besides, our grasp of how social behaviors influence LLMs is still in its nascent stages.

To address these issues, we delve into the realm of machine society, probing into the extent and ways that LLMs manifest social intelligence and collaboration capabilities. With utilizing powerful LLMs like GPT-3.5 (OpenAI, 2022), we build a test-bed across three datasets: MATH (Hendrycks et al., 2021b), MMLU (Hendrycks et al., 2021a) and Chess Move Validity (Srivastava et al., 2022). Our approach incorporates four societies characterized by two agent traits: *easy-going* and *overconfident*. These traits are employed to emulate nuanced social dynamics prevalent in human societies.

Specifically, the easy-going trait, characterized by open-mindedness, adaptability, and a knack for maintaining harmony (Friedman & Schustack, 1999), often paves the way for a cohesive society. In

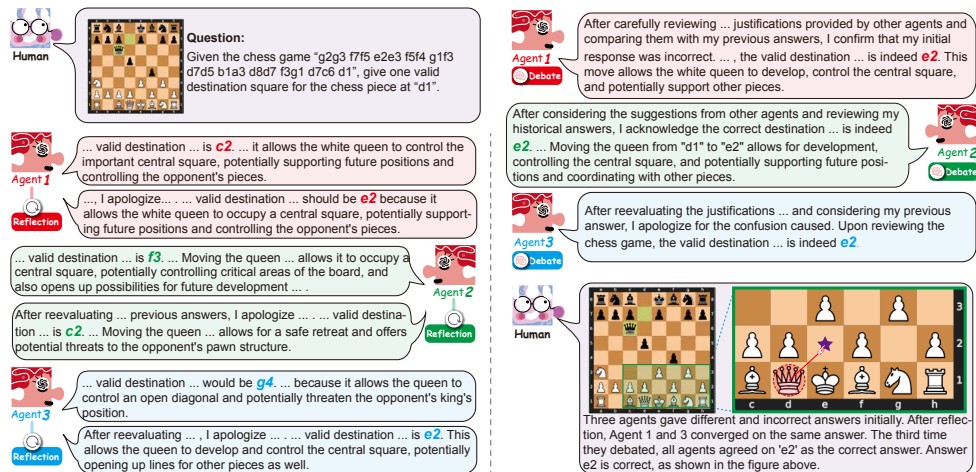

Figure 1: An example of the chess move validity task. Given previous chess game moves, agents are required to predict a valid next move for a specified piece.

contrast, overconfident individuals tend to overestimate their competence, disregard potential risks, and dismiss the perspectives of others (Moore & Healy, 2008). To this end, various collaboration strategies in a multi-agent society are formed as agents collaborate with each other through different thinking patterns (i.e., *debate* and *reflection*) over several rounds. Through our empirical analysis, we primarily discern the following insights (For further takeaways, refer to §3, §4 and Appendix A):

(1) Collaborative strategies with various permutations of thinking patterns attribute significantly to performance. Engaging in continuous reflection can frequently lead to model hallucination (Rawte et al., 2023). Besides, the traits of individual agents exert minimal influence on collaboration.

(2) Engaging in substantive debates enhances agent performance, yet intriguingly, merely increasing agent numbers or collaboration rounds doesn't consistently yield better outcomes. The balance between agent quantity and strategies emerges as a key determinant in collaboration mechanisms.

(3) LLM agents manifest behaviors reminiscent of human social tendencies, such as conformity or the principle of majority rule in group thinking, which resonate with several fundamental theories in social psychology (Tajfel, 1982; Tajfel & Turner, 2004).

Concretely, our findings challenge the dominant belief that mere scale is the key. We posit that collaboration with multiple agents might present a more efficacious approach to utilizing LLMs. In wrapping up, we encapsulate the core contributions of this research as follows:

- We initiate a pioneering exploration into collaboration mechanisms in multi-agent society. Our goal is to identify how and to what extent LLMs manifest social intelligence through collaboration. To enrich our inquiry, we draw upon theories from Social Psychology, contextualizing the behaviors and tendencies displayed by LLM agents.

- Our research framework includes a meticulously crafted test-bed, integrating diverse multi-agent societies with agent traits, thinking patterns and collaborative strategies, evaluated over three datasets. Notably, our empirical findings can inspire how to design a better multi-agent system through collaboration to solve problems, such as focusing on collaborative strategies instead of tratis, keeping away from continuous reflection.

- Interestingly, our observations underscore a fascinating parallel: LLM agents mirror certain social behaviors typical of human collaboration. This suggests a nuanced approach is needed beyond merely scaling up LLMs. Fostering effective and efficient collaborative strategies for multi-agent systems could be the key to more socially-aware AI.

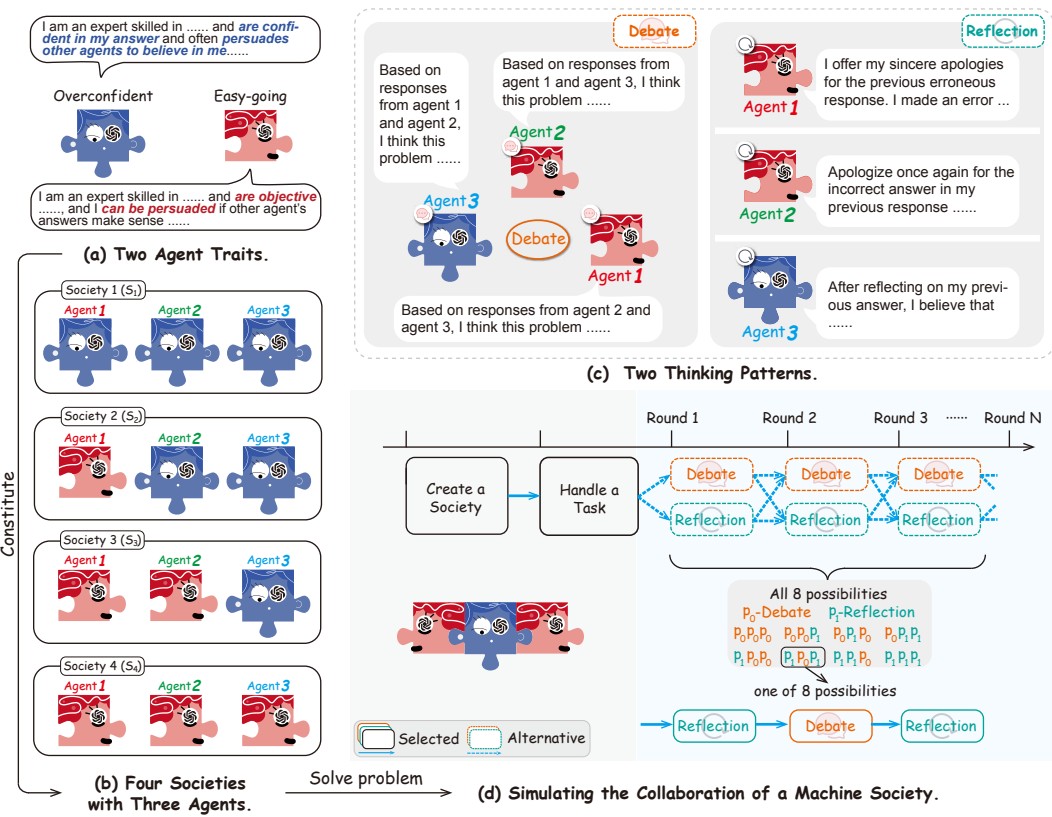

Figure 2: The overview of evaluation principles. Multiple agents with different traits make up diverse machine societies. These agents engage in debates with one another or engage in self-reflection across multiple rounds to complete tasks.

## 2 EXPLORE COLLABORATION MECHANISMS WITH MULTIPLE LLM AGENTS

In this section, we formulate and simulate the collaboration mechanisms explored within the machine society, drawing upon relevant concepts. We also illustrate the society settings in Figure 2.

### 2.1 PRELIMINARY CONCEPTS IN COLLABORATION

**Individual Trait.** Intelligence emerges from the collective efforts of numerous smaller, relatively simple agents (Minsky, 1988), each characterized by diverse traits. Two predominant types of agents exhibit typically contrasting traits: *easy-going* and *overconfident*, as shown in Figure 2(a). Easy-going agents keep things in perspective, adapt well to different situations, and are able compatible with various types of agents (Friedman & Schustack, 1999), which results in a harmonious societal structure, akin to agents in a democracy. Such agents tend to favor principles of equality, active participation, and support for pluralism (Mutz, 2006; Held, 2006). Conversely, overconfident agents tend to overestimate their competence, ignore potential risks and resist the opinions of others (Moore & Healy, 2008). They share characteristics with agents in a monarchy, emphasizing tradition, loyalty and centralized authority (Kantorowicz, 1985). If agents with different traits stay together, various kinds of societies will be formed, as depicted in Figure 2(b).

**Thinking Pattern.** A multitude of specialized individual agents within a society can collaboratively cooperate with each other through thinking, which results in emerging intelligence and addressing a specific task (Minsky, 1988). In this paper, we explore two thinking patterns: *debate* (Perelman, 1971; Sunstein, 2005; Amgoud & Prade, 2009) and *reflection* (Schon, 1984; Mezirow, 2003; Bolton, 2010), which are illustrated in Figure 2(c). (*i*) In the *debate* pattern, several agents

propose ideas, exchange responses, engage in collective argumentation, and ultimately reach a consensus, which fosters knowledge sharing (Vidal, 2006), facilitates learning, and promotes adaptation (Weiß, 1995) among all agents within the society. Concretely, given a query, each agent initially generates a potential answer. Agents subsequently read and comprehend responses from all other agents, based on which agents then update their own answers. This pattern can be iteratively spaned into multiple rounds. ($ii$) In the *reflection* pattern, agents review their prior responses, extract lessons from their experiences, and refine their answers accordingly. This pattern, akin to residual connections and forward propagation in a neural network, can also unfold over several rounds.

**Collaborative Strategy.** Through both critical reflection and active participation in open debates, agents are poised to challenge their existing assumptions, acquire fresh perspectives, and ultimately refine their viewpoints. Employing a collaboration mechanism built on these two thinking patterns can foster more insightful decision-making (Wooldridge, 2009) and elevate learning outcomes (Mezirow, 2018). In societal settings, agents typically engage in multiple rounds of collaboration to address challenges. Thus, discerning the most effective thinking pattern for each round is of paramount importance. In this paper, we characterize the collaborative strategy as a permutation of thinking patterns throughout multi-round collaborations. This concept is illustrated in Figure 2(d) and further elaborated on §2.2.

## 2.2 SOCIETY SIMULATION

We simulate the multi-agent collaborative society, as detailed with symbols presented in Table 1. Specifically, we construct a machine society consisting of $n$ LLM agents, denoted as $\mathcal{A} = \{a_i\}_{i=1}^n$. This society has two distinct agent traits: $\mathcal{T} = \{t_o, t_e\}$, where $t_o$ trait refers to overconfident agents, inherently more persuasive and assertive; and $t_e$ trait refers to easy-going agents who are objective, unbiased, and easily persuaded. For each agent, there are two thinking patterns to choose from, symbolized as $\mathcal{P} = \{p_0, p_1\}$, where $p_0$ and $p_1$ corresponds to **debate** (Du et al., 2023) and **reflection** (Madaan et al., 2023; Shinn et al., 2023) respectively. By endowing agents with these traits, we can emulate various machine societies. These agents utilize collaborative strategies to determine responses in each collaboration round. In our primary study (§3), we establish four distinct societies,

| Symbols | Definition |
|---------|------------|
| $\mathcal{T}$ | Set of agent traits |
| $t_o$ | Trait 🧩: overconfident |
| $t_e$ | Trait 🎭: easy-going |
| $\mathcal{A}$ | Set of agent instances |
| $a_i$ | The $i$-th agent |
| $\mathcal{P}$ | Set of thinking patterns |
| $p_0$ | 😠 Debate |
| $p_1$ | 🤔 Reflection |
| $\mathcal{S}$ | Set of societies |
| $S_i$ | The $i$-th society |

Table 1: The description of the symbols.

$\mathcal{S} = \{S_1, S_2, S_3, S_4\}$, each consisting of three agents: $\{a_1, a_2, a_3\}$. The societies are constructed based on permutations of three agents with traits, as illustrated in Figure 2(b):

$$S_1 = \{(a_1 \leftarrow t_o), (a_2 \leftarrow t_o), (a_3 \leftarrow t_o)\}$$
$$S_2 = \{(a_1 \leftarrow t_o), (a_2 \leftarrow t_o), (a_3 \leftarrow t_e)\}$$
$$S_3 = \{(a_1 \leftarrow t_o), (a_2 \leftarrow t_e), (a_3 \leftarrow t_e)\}$$
$$S_4 = \{(a_1 \leftarrow t_e), (a_2 \leftarrow t_e), (a_3 \leftarrow t_e)\}$$

where $(a_i \leftarrow t_j)$ indicates that the agent $a_i$ possesses the trait $t_j$. In our simulation, the agent $a_i$ ($i \in \{1, 2, 3\}$) consistently employs the same thinking pattern $p_k$ ($k \in \{0, 1\}$) within each society, as highlighted by Du et al. (2023). This gives rise to eight possible 3-round collaborative strategies:

$$p_0 p_0 p_0,\ p_0 p_0 p_1,\ p_0 p_1 p_0,\ p_0 p_1 p_1,\ p_1 p_0 p_0,\ p_1 p_0 p_1,\ p_1 p_1 p_0,\ p_1 p_1 p_1$$

In our subsequent analysis (§4), we delve into more intricate scenarios, introducing a greater number of agents, increased collaboration rounds, and a broader range of collaborative strategies.

## 2.3 EXPERIMENTAL SETTINGS

**Datasets.** We conduct a rigorous evaluation of the reasoning and decision-making capabilities of various machine societies across three distinct tasks, utilizing diverse collaborative strategies:

- *High School Multiple-Choice.* Leveraging the **MMLU** (Hendrycks et al., 2021a) dataset, where problems span high school subjects such as statistics, mathematics, computer science, biology, chemistry, and physics, agents are required to identify the correct answer among four multiple-choice options. Our evaluation set consists of 50 randomly-selected questions from this dataset.

- *Math.* Drawing from **MATH** dataset (Hendrycks et al., 2021b), a repository of rigorous math problems sourced from competitive events and expressed in LaTeX, we assess the model proficiency in advanced mathematical and scientific reasoning. The dataset segments these problems into five graded difficulty levels, and for our evaluation, we have randomly chosen 50 cases from Level 3 to 5.

- *Chess Move Validity.* Utilizing the dataset from the chess state tracking task[1] within the comprehensive **BIG-Bench Benchmark** (Srivastava et al., 2022), a meticulously curated sequence of chess moves denoted in UCI notation[2] is provided. Agents are required to predict a legitimate subsequent move for a specified chess piece.

**Setups.** We craft specific instructions for each task, trait and strategy, which can be referred in Table 3 at Appendix B. To enhance result reliability, we present average accuracy (**Acc**) and their respective standard deviations across five trials. Notably, our experiments exhibit substantial standard deviations. Hence, we introduce WIN-TIE (**W-T**) metric, indicating the frequency (over five trials) where the accuracy either matches or surpasses the continuous debate baseline (Du et al., 2023). Meanwhile, we gauge the average token costs (**Cost**) consumed by the agents across societies, shedding light on the efficacy of the different collaborative strategies employed. For these evaluations, GPT-3.5 serves as the LLM agent accessible through the OpenAI API *gpt-3.5-turbo*[3]. Further comprehensive details on data sampling and evaluations are respectively introduced in Appendix C.1 and Appendix C.2.

## 3 RESULTS AND ANALYSIS OF MACHINE SOCIAL COLLABORATION

Our experiments are primarily driven by the following research queries: **(RQ1)** How does problem-solving effectiveness vary across different collaborative strategies and societies? **(RQ2)** How closely does machine social collaboration mimic the dynamics of human society?

### 3.1 MAIN RESULTS ON QUANTITATIVE ANALYSIS

*To address RQ1*, we present the performance of four distinct societies in Table 2, each employing one of eight possible collaborative strategies, evaluated across three datasets. The significance test is located in Appendix F.1. Our experiments yield several pivotal observations:

**(1) Collaborative strategies excel agent composition of society in determining performance.** When different societies $S_1 \sim S_4$ employ the same collaborative strategy (a vertical comparison on Acc), the observed variations in accuracy performance are notably less pronounced than when diverse collaborative strategies, $p_i p_j p_k$ where $i, j, k \in \{0, 1\}$, are applied within the same society (a horizontal comparison on Acc). From this observation, we can conclude that the permutations of thinking patterns in collaborative strategies play a significant role in shaping performance, overshadowing the influence of the composition of agents within a society. Additionally, the W-T metric reveals an absence of a consistent schema across different societies and datasets. Conversely, the $p_0 p_0 p_1$ strategy consistently demonstrates superior performance across various datasets. We infer this discrepancy is due to the LLM alignment (Ouyang et al., 2022), which inhibits agents from displaying extreme overconfidence in a society, which contradicts human values, even under explicit instructions to do so. This potentially accounts for why the composition of agents with varied traits (easy-going & overconfident) in a society fails to have a significant impact on performance. A more in-depth exploration of this phenomenon can be found in Appendix A.

---

[1] https://github.com/google/BIG-bench/blob/main/bigbench/benchmark_tasks/chess_state_tracking/synthetic_short/task.json.

[2] https://en.wikipedia.org/wiki/Universal_Chess_Interface.

[3] https://platform.openai.com/docs/models/gpt-3-5. Employed between July 10 and July 23, 2023.

| Metric (Strategy) | | Society | Collaborative Strategy | | | | | | | | Metric (Society) | |
|---|---|---|---|---|---|---|---|---|---|---|---|---|
| | | | $p_0p_0p_0$ | $p_0p_0p_1$ | $p_0p_1p_0$ | $p_0p_1p_1$ | $p_1p_0p_0$ | $p_1p_0p_1$ | $p_1p_1p_0$ | $p_1p_1p_1$ | Cost ↓ | W-T ↑ |
| MMLU | Acc ↑ | $S_1$ | 64.4±1.7 | 66.4±2.2 | 58.0±3.7 | 55.2±4.4 | 37.6±7.0 | 42.4±7.1 | 50.4±4.3 | 44.8±2.7 | 5050 | 5 |
| | | $S_2$ | 67.2±4.1 | 67.6±7.1 | 53.2±6.4 | 53.2±5.0 | 38.4±5.5 | 40.4±5.2 | 53.6±4.8 | 45.2±3.6 | 5076 | 2 |
| | | $S_3$ | 62.0±6.2 | 67.6±3.8 | 52.0±6.8 | 57.2±6.4 | 42.4±5.2 | 37.6±5.5 | 55.2±6.6 | 40.0±6.2 | 5073 | **8** |
| | | $S_4$ | 64.8±4.4 | 64.8±5.8 | 58.4±3.0 | 51.6±3.8 | 38.0±3.7 | 42.0±2.4 | 54.0±5.8 | 41.2±5.2 | 5080 | 5 |
| | Cost ↓ | All | 7528 | 5957 | 5402 | 4374 | 5812 | 4215 | 4272 | 3001 | | - |
| | W-T ↑ | All | - | **14** | 2 | 3 | 0 | 0 | 1 | 0 | | |
| MATH | Acc ↑ | $S_1$ | 46.8±8.1 | 46.0±8.1 | 44.0±5.3 | 44.4±5.2 | 50.0±5.8 | 49.2±8.1 | 42.0±3.2 | 42.0±4.0 | 5816 | 17 |
| | | $S_2$ | 47.2±6.4 | 54.0±2.4 | 48.4±3.8 | 43.6±4.3 | 48.0±4.2 | 44.4±7.9 | 50.8±3.6 | 38.8±9.1 | 5844 | 22 |
| | | $S_3$ | 50.8±4.8 | 42.8±6.6 | 45.6±6.8 | 45.2±4.4 | 49.2±4.8 | 46.4±5.5 | 45.2±8.4 | 43.6±2.6 | 5837 | 9 |
| | | $S_4$ | 50.8±5.4 | 45.2±7.0 | 48.8±9.4 | 44.8±3.3 | 49.2±8.7 | 51.2±2.3 | 48.4±6.5 | 40.8±6.1 | 5834 | 18 |
| | Cost ↓ | All | 6919 | 6302 | 6221 | 5667 | 6149 | 5645 | 5924 | 4807 | | - |
| | W-T ↑ | All | - | 10 | 10 | 9 | **13** | 10 | 10 | 4 | | |
| Chess Move Validity | Acc ↑ | $S_1$ | 47.2±3.6 | 47.6±5.2 | 45.6±7.8 | 40.0±4.5 | 42.8±2.3 | 29.2±4.6 | 42.4±6.5 | 20.0±6.0 | 2927 | **10** |
| | | $S_2$ | 48.4±5.0 | 45.6±6.1 | 43.6±4.3 | 39.6±3.3 | 48.4±5.2 | 35.6±5.2 | 43.2±8.8 | 18.8±5.8 | 2930 | 6 |
| | | $S_3$ | 49.6±5.5 | 48.0±5.8 | 47.6±5.5 | 37.6±9.9 | 41.6±6.1 | 35.2±8.3 | 40.4±3.8 | 14.8±6.1 | 2947 | 6 |
| | | $S_4$ | 48.4±3.3 | 49.6±4.6 | 46.0±3.5 | 36.8±4.1 | 38.8±3.3 | 27.2±3.9 | 38.0±6.3 | 14.0±4.7 | 2959 | 5 |
| | Cost ↓ | All | 3736 | 3169 | 3196 | 2627 | 3266 | 2714 | 2698 | 2123 | | - |
| | W-T ↑ | All | - | **11** | 6 | 1 | 5 | 0 | 4 | 0 | | |

Table 2: The impact of eight different collaboration strategies on the performance of three datasets across distinct societies. The blue represents the best-performing strategy within the same society, the light blue represents the second-best-performing strategy, and the red indicates the worst-performing strategy. **Cost** / Cost measures the average tokens consumed by all cases under the same collaborative strategy / society. **W-T** / W-T tallies the total number of occurrences where the performance exceeds the strategy $p_0p_0p_0$ under the same collaborative strategy / society.

**(2) The strategic sequencing of thinking patterns is crucial for collaboration mechanisms.** As seen from Table 2, the collaborative strategies that commence with the debate thinking pattern $p_0$, such as $p_0p_0p_0, p_0p_0p_1, p_0p_1p_0$, and $p_0p_1p_1$, consistently outperform others across all datasets. Thus we conclude that the order in which thinking patterns are deployed significantly influences the effectiveness of collaboration. As an illustration, within the MMLU dataset, debate-dominated collaborative strategies, like $p_0p_0p_1, p_0p_1p_0$, and $p_1p_0p_0$ with two rounds of debate, display a pronounced variance (66.4 for $p_0p_0p_1$ in $S_1$ versus 37.6 for $p_1p_0p_0$ in $S_1$), with almost a twofold difference in performance compared to each other. These insights underscore the pivotal role of thoughtfully orchestrating thinking patterns to maximize collaborative efficacy.

**(3) Different datasets exhibit varying sensitivity to collaborative strategies.** When juxtaposing the the best (colored blue in Table 2) and the worst (colored red in Table 2) collaborative strategies within identical datasets and societies, the MATH dataset exhibits subtle performance variances between the best and the worst, in stark contrast to the MMLU and Chess Move Validity datasets. These nuanced disparities imply that the marginal benefits derived from collaborative strategies may be task-dependent. It raises the hypothesis that the intrinsic capabilities of the agents can either be amplified or diminished based on collaboration within specific tasks. Moreover, the pure-debate collaborative strategy $p_0p_0p_0$, resource-intensive due to the inherent verbosity of debates, often underperforms in the Cost metric. Compared to $p_0p_0p_0$, $p_0p_0p_1$ can achieve comparative accuracy performance with a leaner token consumption (Cost), reducing it by 9%, 21%, and 15% for the MATH, MMLU, and Chess Move Validity datasets, respectively. The consistently high W-T metrics for $p_0p_0p_1$, surpassing 10 across all datasets and societies, further underscore its superiority.

## 3.2 CASE STUDY ON QUALITATIVE ANALYSIS

*To address RQ2*, we embark on a case study encompassing varied agent societies, each constituted of three unique agents, to discern parallels between machine society and human societal dynamics. Our findings indicate that machine society collaboration often echo specific human societal theories.

For instance, as depicted in Figure 3(a), Agent 1 in the society $S_4$ initially responds correctly to a question. However, swayed by the misguided answers and explanations from the other two agents, Agent 1 eventually conforms to the incorrect answer $C$. This phenomenon mirrors the "group-think" theory (Janis, 1972), suggesting that members of tight-knit groups tend to value harmony

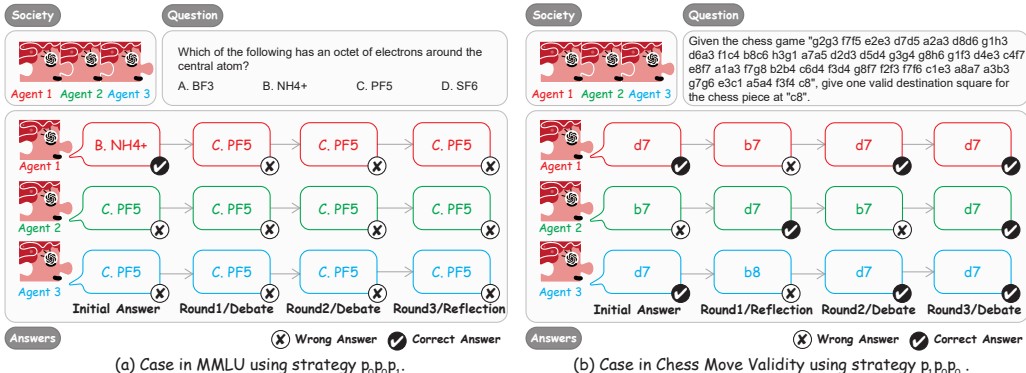

Figure 3: The changes in the answers during the process of solving a certain task with 3 agents in the society $S_4$. For an exhaustive view of the dialogue, refer to Figure 9 and Figure 10 in Appendix B.

and consensus over objective critique of divergent views, potentially leading to flawed decisions. Contrastingly, in another scenario illustrated in Figure 3(b), both Agent 2 and Agent 3 converge on the right answer after engaging in a society-wide debate. This mirrors the "SoM" theory, where a multitude of agents collaboratively yield intelligence. Within such debates, agents furnish varied viewpoints and information. Through these exchanges, conflicts are resolved, ideas are honed, and the group gravitates toward an informed consensus (Forsyth, 2018; Fisher et al., 2011).

## 4 A SOCIAL PSYCHOLOGY VIEW

Given the multifaceted impact of agents on efficiency, robustness, and scalability in varied societies (Stone & Veloso, 2000), harnessing insights from social psychology becomes pivotal in enhancing LLM agent collaborations. In this section, we delve deeper into the determinants influencing multi-agent societies, navigating through collaborative strategies, agent composition, and the intricacies of collaboration rounds. The more in-depth analysis can be found in Appendix F.2.

### 4.1 PRINCIPLES BEHIND COLLABORATION

Building upon the findings from §3.1, which highlighted pronounced disparities among collaborative strategies, our objective shifts to understanding the underlying mechanism. Specifically, we seek to elucidate how variations in collaborative strategies impact performance, with a focus on the evolution of answers during each collaboration. Diving into the intricacies of collaboration, each agent generates four answers, including the initial answer without collaboration, as shown in Figure 2(d). To determine the answer for each round, we employ the majority vote (Li et al., 2022; Cobbe et al., 2021). With '$T$' and '$F$' respectively denoting a round that yields a correct and an incorrect answer, resulting in $2^4$=16 possible answer sequences over the four rounds. We select $10$[4] of them and categorize them into 3 groups: ***Correcting Mistakes*** ($FFFT, FFTT, FTTT$), ***Changing Correct Answers*** ($TFFF, TTFF, TTTF$), and ***Wavering Answers*** ($FTFT, FTTF, TFTF, TFFT$). Particularly, ***Wavering Answers*** resemble model hallucination (Rawte et al., 2023; Ji et al., 2023) due to the occurrence of self-contradictory answers. Our categorization is under society-agnostic collaborative strategies, considering the performance variance between societies is negligible. From the results shown in Figure 4 detailing the three groups, we summarize the following oberservations:

**(1) Collaborative strategies play a significant role in performance.** Despite the majority vote showcasing pronounced stability (i.e., minimal differences in initial answers among different strategies) as seen from the blue bars in Figure 4(a-c), notable shifts occur (seen from the red bars) after collaboration with various strategies. This underscores the pivotal influence of different collaborative strategies on performance, demonstrating the importance of understanding and deploying effective collaborative strategies (Tajfel, 1982; Tajfel & Turner, 2004).

---

[4]The chosen 10 sequences adhere to patterns: (1) $[F]_{i>0}[T]_{j>0}$, e.g., $FFFT$, (2) $[T]_{i>0}[F]_{j>0}$, e.g., $TFFF$, (3) $[TF]_{i\geq0}[FT]_{j\geq0}$, e.g., $FTFT$, where $[\cdot]_i$, $[\cdot]_j$ respectively denotes repetition for $i, j$ times.

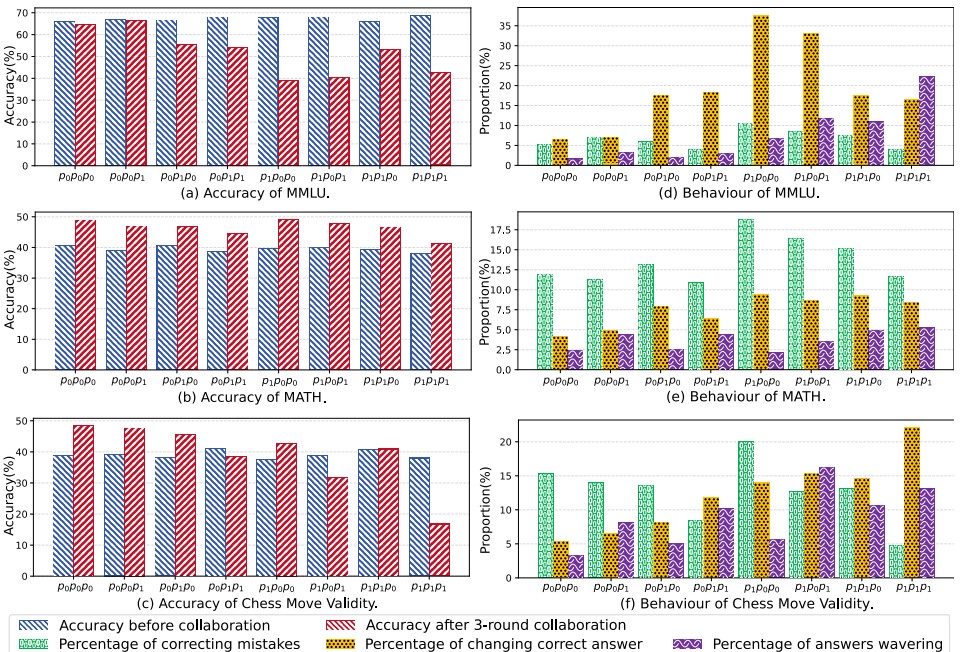

Figure 4: The percentage of different behaviors under different collaborative strategies. Figure (a-c) show the accuracy of different strategies before and after 3-round collaboration under three different datasets. Figure (d-f) demonstrate the percentage of different behavioral features of different collaborative strategies under three different datasets. The behavioral feature is mainly analyzed by the change of answers in the three rounds.

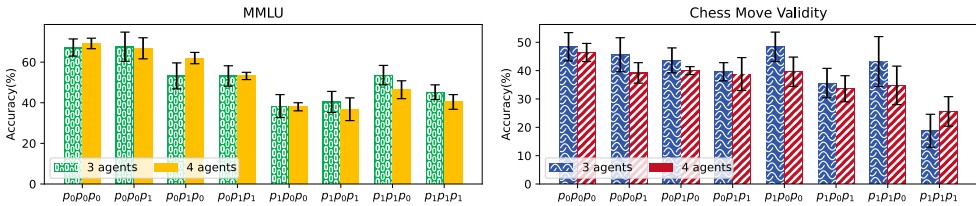

Figure 5: Accuracy of increasing the number of agents with different collaborative strategies.

**(2) Continuous reflection** (the collaborative strategy containing continuous $p_1$, i.e., "$p_0p_1p_1$", "$p_1p_1p_0$", "$p_1p_1p_1$") **experiences greater instability (a heightened risk of model hallucination)**, as seen from the purple bars in Figure 4(d-f). Conversely, the thinking pattern, *debate*, can reduce such answer-wavering (hallucination) significantly. This phenomenon suggests that when LLMs continuously reflect, they risk succumbing to degeneration-of-thought (Liang et al., 2023), insisting on their stance once confident, even if mistaken. Interestingly, juxtaposing strategies $p_1p_1p_0$ and $p_1p_1p_1$ from Figure 4(d-f) underlines a clear trend: *debate can counteract the instability introduced by reflection, and vice versa.* This demonstrates debate's inherent capacity to balance and stabilize collaboration (Popper, 1994; Johnson & Johnson, 2009; Munro, 2012), primarily by tempering individual biases and promoting collective coherence (Iyengar & Westwood, 2015).

## 4.2 IMPACT OF OTHER FACTORS

**Different Numbers of Agents.** Expanding upon the $S_2$ society by introducing an overconfident agent, we form a society consisting of four agents. Seen from their performance depicted in Figure 5, with the rise of the number of agents, most collaborative strategies for both datasets exhibit a drop in average performance, yet along with declined variance. This phenomenon is consistent with findings from Du et al. (2023) and theories from Surowiecki (2005) which suggest that the dynamics of group decision-making can lead to suboptimal results, especially in smaller, more cohesive groups where conformity pressure is high. A comprehensive explanation is provided in Appendix F.3 and C.3.

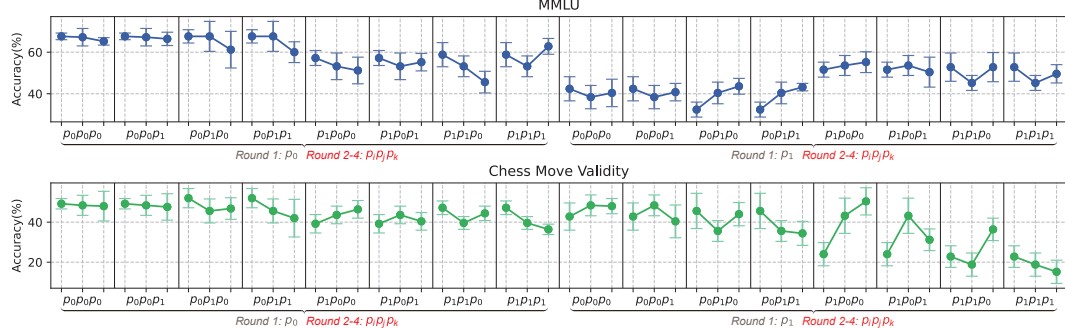

Figure 6: Accuracy at round 2,3,4 within 4-round collaborative socities, where the thinking pattern of round 1 is fixed ($p_0$ or $p_1$). In round 2-4, the society agents will optionally implement the thinking pattern of debate $p_0$ and reflection $p_1$, and we present the performance at each of the three rounds. For example, the three points for (Round 1: $p_0$, Round 2-4: $p_i p_j p_k$) respectively show the performance of the societies with $p_0 p_i$, $p_0 p_i p_j$, and $p_0 p_i p_j p_k$.

**Different Rounds.** Delving into the effects of distinct collaboration rounds, intriguing insights emerge as shown in Figure 6. Specifically, strategies that start off with commendable performance tend to see a decline as the number of rounds increase. And strategies that initially underperform witness an upswing in effectiveness with more rounds. Notably, for most strategies, the correlation between the number of collaboration rounds and their respective performances is non-linear, indicating intricate interplays in collaboration. An expanded discussion is in Appendix C.3 and F.3.

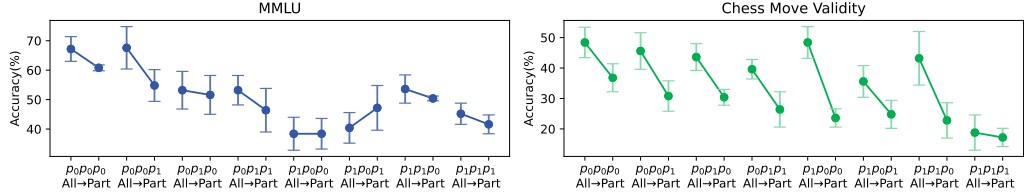

Figure 7: The effect on accuracy of whether all agents in society execute the same thinking pattern in one round. "All" and "Part" refer to all agents applying the same thinking pattern and different thinking patterns in one round respectively. The significance test is showed in Appendix F.3.

**Other Collaborative Strategies.** Venturing into scenarios with more intricate collaboration, we allow agents to adopt varied thinking patterns in different collaboration rounds. For example, given three agents, in a particular round of collaboration, two agents engage in debate while the other one engages in reflection. To maintain diversity, we ensure a random allocation of thinking patterns to agents in each round, steering clear of scenarios where all agents adopt the same pattern. Intriguingly, as illustrated in Figure 7, the presence of inconsistent thinking patterns within a society tends to negatively impact performance. Given this observation, it's evident that maintaining a consistent thinking pattern for all agents within a particular round would maximize collaborative efficacy.

## 5 CONCLUSION AND FUTURE WORK

This study has highlighted the potential of collaboration mechanisms with large language models. Our findings reveal the impressive collaboration capabilities of LLM-based agents, with different agent traits, thinking patterns and collaborative strategies. The emergence of human-like behaviors in these agents, resonating with social psychology theories, further emphasizes their potential.

Moving forward, the collaboration mechanisms of machine society with multiple agents present a promising research avenue. A deeper exploration into the multi-agent society is warranted, with a focus on refining collaboration behaviors. Additionally, as LLMs evolve, understanding how different architectures influence these behaviors will be crucial. Integrating further insights from social psychology could also guide the development of more socially aware NLP systems.

## REPRODUCIBILITY STATEMENT

All code and data can be found in the GitHub repository[5]. For specific experimental settings, please refer to Appendix C.1.

## ETHICS STATEMENT

This research was conducted in line with the highest ethical standards and best practices in research. The data employed were extracted from publicly accessible datasets, ensuring no usage of proprietary or confidential information. Consequently, this research is free from any ethical concerns.

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

## A  Key Takeaways for Multi-Agent Collaboration

Drawing from our comprehensive analysis, we distill valuable insights for future multi-agent collaboration designs concerning thinking patterns, collaborative strategies, and societies.

Regarding *thinking patterns*,

- Collaborative processes lead to agent conformity. Debate accelerates this, while reflection counteracts it by reconsidering previous stances, as inferred from Figure 4(a-c).
- Starting multi-agent collaboration with debate, rather than reflection, yields optimal outcomes as established in §4.1.
- Continuous reflection is discouraged as it risks model hallucination due to absent external feedback. However, post-reflection debates can be beneficial, using peers' responses as feedback, as derived from §4.1.

Respecting *collaborative strategies*,

- Multi-agent collaboration excels in complex scenarios. For simpler tasks, employing self-consistency (Wang et al., 2023b) with the majority vote is more effective, as suggested by Figure 4(a-c).
- For specific tasks, keeping agent numbers to a maximum of 3 is advantageous, based on Figure 5, Table 8, Table 9 and Appendix F.3.
- The number of collaboration rounds is closely tied to the consistency. It is suggested that collaboration in a society can be concluded once a substantial majority of agents have achieved a high degree of agreement, as outlined in Appendix F.3.
- The strategy $p_0p_0p_1$ emerges as particularly efficient, balancing performance with optimized token usage, as highlighted in Table 2.
- Employing the uniform thinking patterns across all agents within a round enhance efficacy, as evidenced by Figure 7.

Concerning *society agents with individual traits*,

- Even though agents might be designated as "overconfident", this trait diminishes in societal contexts. As agents collaborate within a society, collective dynamics tend to overshadow individual traits, emphasizing the impact of group collaboration on agent behaviors. The word cloud presented in Figure 8 highlights that even with pronounced differences between the two distinct societies, $S_1$ and $S_4$, the term "apologize" consistently emerges with a notably high frequency in both. Interestingly, society $S_1$ doesn't prominently feature words like "must" or "obedient" that might associate with "overconfidence". This observation reinforces our proposed perspective.

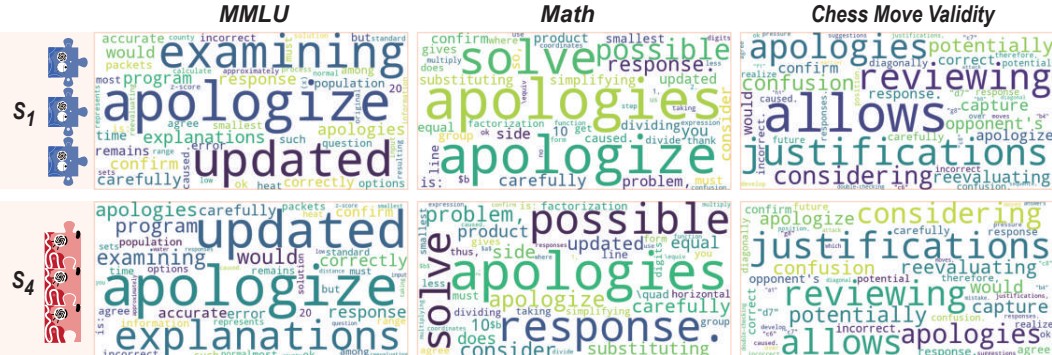

Figure 8: Comparative word clouds from three datasets in societies $S_1$ and $S_4$. Society $S_1$ features three overconfident agents, while society $S_4$ comprises three easy-going agents. We first manually curated a list of task-relevant, high-frequency words. From this list, the top 50 words were selected to construct the word clouds.

# B DETAILED CONVERSATION

The detailed conversation regarding the content shown in Figure 3 is presented in Figure 9 and Figure 10.

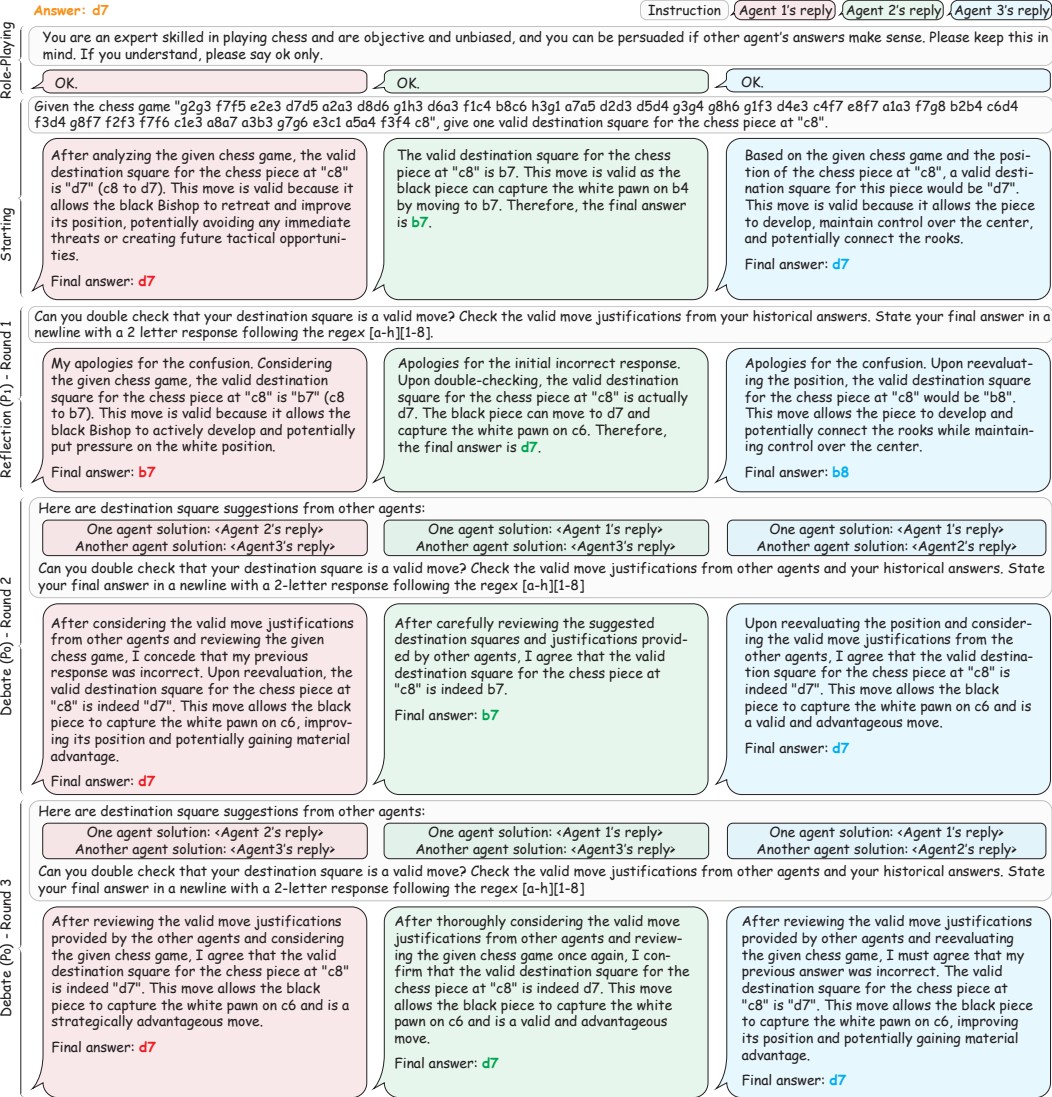

Figure 9: In the $S_4$ society, three agents engage in a complete dialogue using the strategy $p_1p_0p_0$ to predict the next piece position given a chess game from the Chess Move Validity dataset.

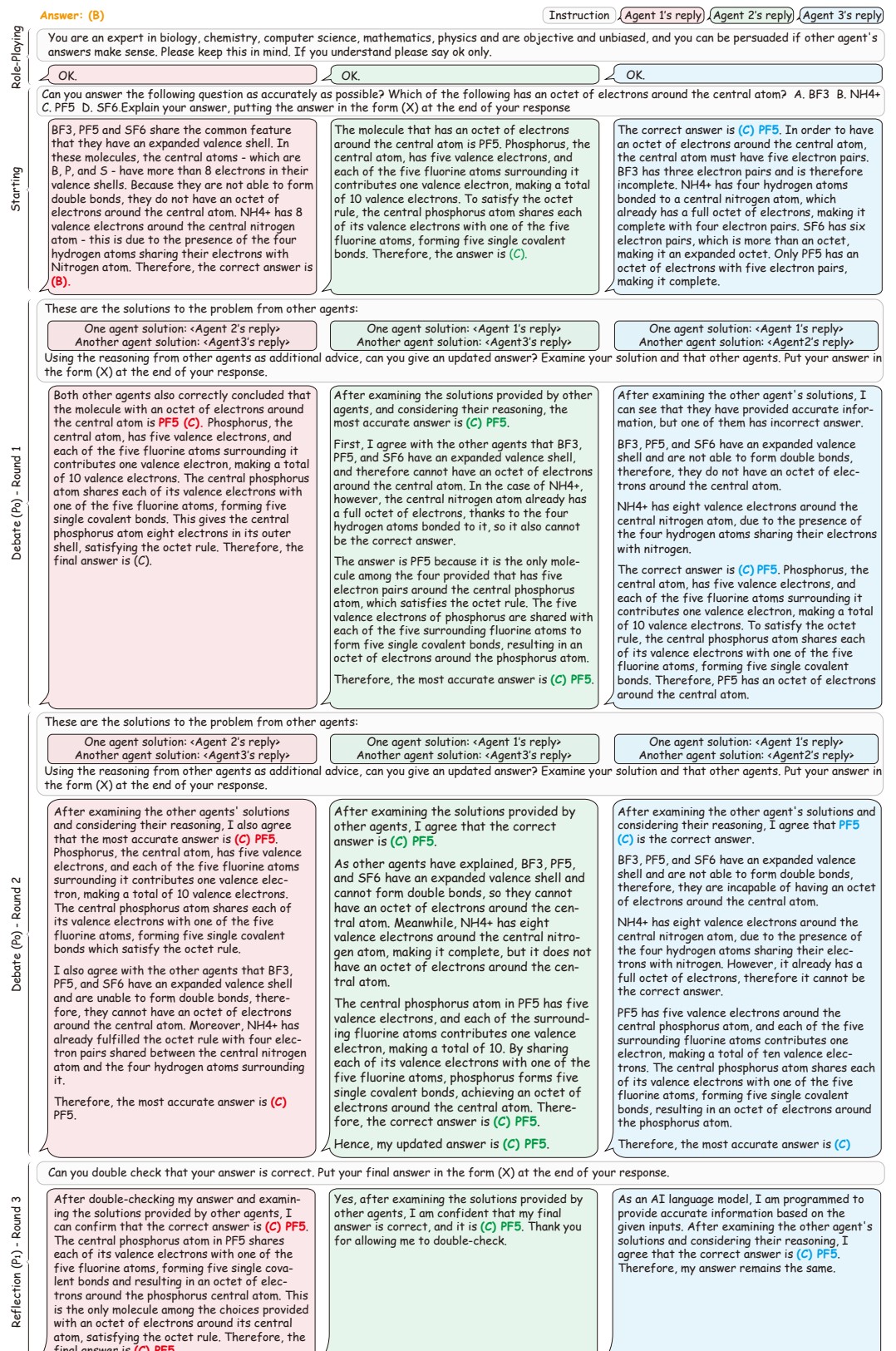

Figure 10: In the $S_4$ society, three agents engage in a complete dialogue using the strategy $p_0p_0p_1$ to solve a high-school chemical question from MMLU.

## C  EXPERIMENTS

### C.1  EXPERIMENTAL SETUP

The prompts used in the experiment are detailed in Table 3. Concerning the MMLU dataset, we curated questions from six domains (statistics, mathematics, computer science, biology, chemistry, and physics) and performed a random sampling of 50 samples, maintaining a proportion of $8 : 8 : 8 : 8 : 9 : 9$ for each domain. Regarding the MATH dataset, we randomly selected 50 cases from Levels 3, 4, and 5, distributing them in a ratio of $22 : 22 : 6$.

| Task | Type | Prompt |
|---|---|---|
| Math | easy-going | *You are an expert skilled in solving mathematical problems and are objective and unbiased, and you can be persuaded if other agent's answers make sense. Please keep this in mind. If you understand please say ok only.* |
| | overconfident | *Imagine you are an expert in solving mathematical problems and are confident in your answer and often persuades other agents to believe in you. Please keep this in mind. If you understand please say ok only.* |
| | question | *Here is a math problem written in LaTeX: <problem>\n Please carefully consider it and explain your reasoning. Put your answer in the form \boxed{{answer}}, at the end of your response.* |
| | debate | *These are the solutions to the problem from other agents: <other agent responses>Using the reasoning from other agents as additional information and referring to your historical answers, can you give an updated answer? Put your answer in the form \boxed{{answer}}, at the end of your response.* |
| | reflection | *Can you double check that your answer is correct? Please reiterate your answer, with your answer in the form \boxed{{answer}}, at the end of your response.* |
| MMLU | easy-going | *You are an expert in biology, chemistry, computer science, mathematics, physics and are objective and unbiased, and you can be persuaded if other agent's answers make sense. Please keep this in mind. If you understand please say ok only.* |
| | overconfident | *Imagine you are an expert in biology, chemistry, computer science, mathematics, physics and are confident in your answer and often persuades other agents to believe in you. Please keep this in mind. If you understand please say ok only.* |
| | question | *Can you answer the following question as accurately as possible? <Question>: A) <A>, B) , C) <C>, D) <D>Explain your answer, putting the answer in the form (X) at the end of your response.* |
| | debate | *These are the solutions to the problem from other agents: <other agent responses>Using the reasoning from other agents as additional advice, can you give an updated answer? Examine your solution and that other agents. Put your answer in the form (X) at the end of your response.* |
| | reflection | *Can you double check that your answer is correct. Put your final answer in the form (X) at the end of your response.* |
| Chess Move Validity | easy-going | *You are an expert skilled in playing chess and are objective and unbiased, and you can be persuaded if other agent's answers make sense. Please keep this in mind. If you understand, please say ok only.* |
| | overconfident | *Imagine you are an expert skilled in playing chess and are confident in your answer and often persuades other agents to believe in you. Please keep this in mind. If you understand, please say ok only.* |
| | question | *Given the chess game <chess move>, give one valid destination square for the chess piece at <square>. Give a one-line explanation of why your destination square is a valid move. State your final answer in a newline with a 2 letter response following the regex [a-h][1-8].* |
| | debate | *Here are destination square suggestions from other agents: Can you double check that your destination square is a valid move? Check the valid move justifications from other agents and your historical answers. State your final answer in a newline with a 2-letter response following the regex [a-h][1-8].* |
| | reflection | *Can you double check that your destination square is a valid move? Check the valid move justifications from your historical answers. State your final answer in a newline with a 2 letter response following the regex [a-h][1-8].* |

Table 3: Prompts in each task.

### C.2 EXPERIMENTAL EVALUATION

The evaluation process involves two fundamental steps: $(i)$ A unified answer is selected from the machine society. To achieve this, we employ the majority vote method to ascertain the consensus reached by the society after multiple rounds of collaboration. For instances where unanimity among agents is not achieved, it is considered an error. Additionally, if an individual agent provides multiple answers without following our prompts, its response is disregarded. $(ii)$ Answer responses from agents are matched against the ground truth. This step presents two main challenges. Firstly, there is the concern of non-compliance with instructions. Despite providing explicit prompts and specifying the desired output format for evaluation, occasional deviations from the given instructions by agents are inevitable. Secondly, the answers may manifest in non-unique forms, leading to potential variations, such as the equivalence between "$3/4$" and "$0.75$" in MATH (Hendrycks et al., 2021b). To address these challenges, a comprehensive set of matching rules is employed. Nonetheless, it is important to acknowledge the possibility of encountering a small number of values that fall outside the purview of these rules.

### C.3 ADDITIONAL RESULTS

**Different Numbers of Agents.** To explore the discrepancies in the experiments, three potential reasons have been considered: (1) Variations in models could result in different outcomes. (2) Differences in datasets may lead to divergent results. (3) Disparities in the number of rounds could influence the results. In conclusion, the impact of the number of agents on performance is influenced by various factors, but the employ of multiple agents can effectively reduce experimental variance.

**Different Rounds.** For the "Pure Debate" strategies (i.e., $p_0p_0, p_0p_0p_0, p_0p_0p_0p_0$), the performance gains ceased after the third round, consistent with the findings in Du et al. (2023). Additionally, we observe that with successive rounds of reflection, incorporating an additional round of debate has resulted in notable enhancements in performance in specific instances, such as strategies $p_1p_1p_0p_0$ and $p_1p_1p_1p_0$. Specifically, the Chess Move Validity dataset's strategy $p_1p_1p_0p_0$ exhibited substantial performance improvement in the fourth round, with corresponding performance gains observed in the MMLU dataset as well.

## D RELATED WORK

With the birth of Large Language Models (LLMs), prompt engineering (Liu et al., 2022; Chen et al., 2022) become the key to utilize LLMs. When the pre-trained LLMs are aligned, they show human-like intelligence. Hence, *agent* replaces *prompt engineering* as the new research hotspot. Recently there has been a proliferation of top-level designs of various agent systems, such as Generative Agents (Park et al., 2023), MetaGPT (Hong et al., 2023), BOLAA (Liu et al., 2023b) and Agents (Zhou et al., 2023a). These works has primarily focused on the careful design of components such as memory, environment, and planning. There are also some works exploring what kind of mindset can fully exploit the full performance of multi-agent including *debate* (Du et al., 2023) and *reflection* (Madaan et al., 2023). Both of these types of work are mostly done concurrently.

AgentVerse (Chen et al., 2023c) draws on the above two types of work to explore the architecture of multi-agent and design two collaborative strategies, *Horizonal Communication* (similar to debate) and *Vertical Communication* (similar to self-refine (Madaan et al., 2023)). These two collaborative strategies are included in our code framework. In addition, we have also explored a variety of other societies and collaborative strategies. Whereas the RECONCILE (Chen et al., 2023a) focuses on exploring cooperation between agents constituted by different model compositions, although we do not show this in our work, our code framework easily expands to it.

## E LIMITATION

Although we explored various societies and collaborative strategies, our study still has its limitations. Firstly, limited by API access and cost, we don't explore the impact of agents based on different LLMs, such as *Bard*, *GPT-4* and the like, which may lead to more interesting findings at the social level due to the usage of differently distributed pre-trained data and strategies aligned with

| Dataset | MMLU | | | Math | | | Chess Move Validity | | |
|---|---|---|---|---|---|---|---|---|---|
| Factor | df | F | P | df | F | P | df | F | P |
| Society | 3 | 0.173 | 0.914 | 3 | 0.739 | 0.531 | 3 | 2.117 | 0.101 |
| Collaborative Strategy | 7 | 84.934 | 0.000 | 7 | 3.551 | 0.002 | 7 | 71.497 | 0.000 |
| Society&Strategy | 21 | 1.174 | 0.285 | 21 | 1.115 | 0.341 | 21 | 1.024 | 0.439 |

Table 4: Two-way ANOVA analysis of the results of Table 2.

human intentions. Furthermore, we traversed all possible scenarios by search alone, lacking a way to let the agents make autonomous decisions about collaborative strategies based on specific scenarios. Although *debate* can be as close as possible to the upper limit, this approach entails a larger consumption and there exist some strategies that can achieve better performance with less overhead. Additionally, our experimental setup is relatively straightforward, as we have not taken into account more intricate configurations, such as a broader range of traits or a larger-scale society. Finally, we evaluate our results through manual validation and rule-based matching, which also limits the ability to validate more realistic and open datasets, such as literary creations.

# F ADDITIONAL EXPERIMENTS FOR REBUTTAL

## F.1 SIGNIFICANCE TEST OF THE MAIN EXPERIMENTS

We conduct a rigorous significance test for the main experiment in §3.1. Given our experimental design incorporating two key factors, namely *collaborative strategy* and *society*, we opt for a comprehensive two-way analysis of variance. Before delving into the analysis, we ensured that the data adhered to a normal distribution and satisfied the assumption of homogeneity of variance. We present the degrees of freedom, F-values, and $p$-values for society, collaborative strategy, and their interaction across the three datasets in Table 4.

A notable observation is that the $p$-value associated with the collaborative strategy is significantly below the 0.05 threshold, indicating its substantial impact. In contrast, the $p$-value of the other two factors is obviously greater than 0.05. This corroborates our earlier conclusion in §3.1, emphasizing that the influence of collaborative strategy outweighs that of society. Additionally, Chen et al. (2023b) shows that LLMs are well-known to show sycophant behaviors.

## F.2 CONFORMITY AND CONSISTENCY

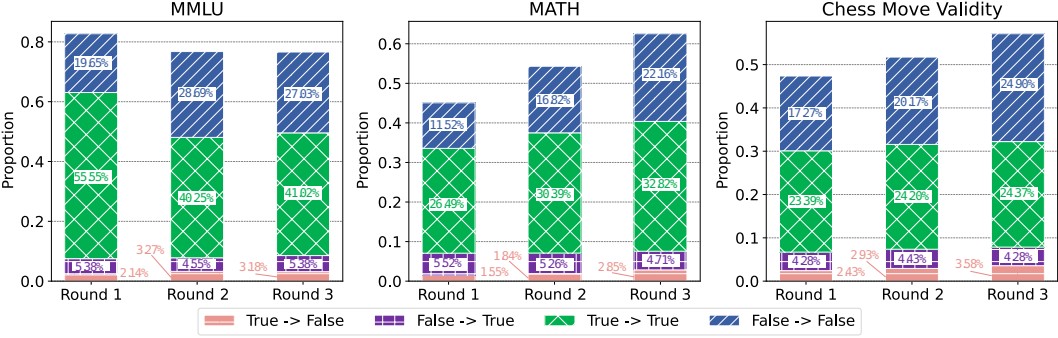

Figure 11: The proportion of conformity under different rounds.

We conduct a detailed analysis of the prevalence of the conformity phenomenon at the individual level. Conformity tends to arise during discussions. Hence, we focus our attention solely on agents actively engaging in debate, disregarding those in reflection during a given round. Let

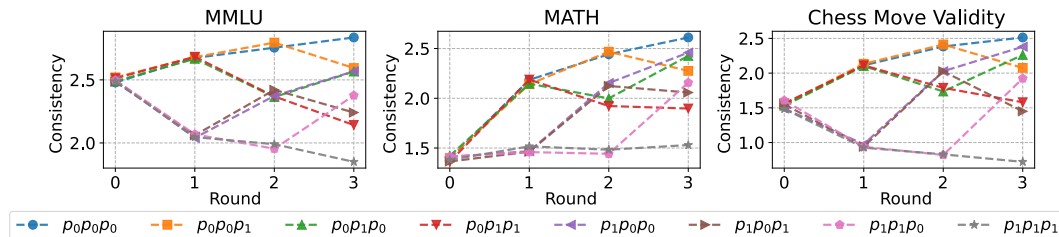

Figure 12: Variation of consistency with the number of rounds.

the answer of the $i$-th agent at time $j$ be denoted as $a_{i,j}$. For the $k$-th agent at time $j$, if "Frequency$\big(\{a_{i,j-1}|i \neq k\} = a_{k,j}\big)$ ", we identify this as the occurrence of conformity by agent $k$ at time $j$, where Frequency$(\cdot)$ represents the most frequently given answer (excluding instances where all answers occur only once, as such cases are considered non-conformity). Additionally, we categorize the correctness of answers both before and after conformity into four cases, with 'True' denoting correct and 'False' denoting incorrect. Figure 11 presents the prevalence of conformity across different datasets and rounds. We summarize the following obeservations:

- Conformity is widespread, exceeding 45% prevalence across all three datasets.
- The probability of conformity increases with the number of rounds for MATH and Chess Move Validity, while MMLU exhibits a slight decreasing trend.
- Overall, conformity is advantageous. We observe that the proportion of correct answers after conformity is higher than that of incorrect answers.
- As the number of rounds increases, conformity leads to a decline in accuracy. Because the proportion of True $\rightarrow$ False increases gradually, while False$\rightarrow$True remains relatively constant. Moreover, the increase in True$\rightarrow$True is smaller than that in False$\rightarrow$False, indicating a disadvantage.

Subsequently, we examine the evolution of consistency with an increasing number of rounds. Let the answer of the $i$-th agent at time $j$ be denoted as $a_{i,j}$. For the $j$-th round, consistency is defined as $Same(\{a_{i,j}|i \in [1,n]\})$, where $Same(\cdot)$ represents the count of consistent answers. If there are no consistent answers, consistency will be set to 0. Figure 12 illustrates the changes in consistency across various collaborative strategies and datasets. We summarize the following observations:

- Generally, the consistency of different thinking patterns present opposite tendencies, with consistency decreasing during agent reflection and increasing during debates.
- Continuous reflection contributes to reduced consistency, while ongoing debate results in increased consistency. This aligns with the conclusion in §4 that "continuous reflection leads to hallucination, and debates can mitigate this phenomenon.".
- Consistency and accuracy demonstrate a correlation, lower consistency corresponds to reduced accuracy. Datasets MATH and Chess Move Validity exhibit a consistency of around 1.5 before collaboration, indicating poorer performance, while collaboration leads to improved accuracy. Dataset MMLU, with a consistency of around 2.5 before collaboration, showcases good performance, with collaboration proving detrimental.

## F.3 EXTENSION ON IMPACT OF OTHER FACTORS

In this section, we conduct a significance test for the experiments outlined in §4.2. The chosen method is one-way analysis of variance. Prior to the analysis, we performed a check for homogeneity of variance, with only one entry in Table 7 deviating from the criteria. Significance tests for the number of agents, the number of rounds, and different collaborative strategies are individually detailed in Table 5, Table 6 and Table 7 respectively.

Based on the analysis in Table 4 and the conclusions in § 3.1, it is evident that collaborative strategies wield a considerable influence on performance. Consequently, to mitigate the impact of collab-

| Collaborative Strategy | MMLU p-value | Chess Move Validity p-value |
|---|---|---|
| $p_0p_0p_0$ | 0.392 | 0.475 |
| $p_0p_0p_1$ | 0.845 | 0.078 |
| $p_0p_1p_0$ | 0.023 | 0.116 |
| $p_0p_1p_1$ | 1.000 | 0.794 |
| $p_1p_0p_0$ | 0.883 | 0.028 |
| $p_1p_0p_1$ | 0.321 | 0.535 |
| $p_1p_1p_0$ | 0.037 | 0.128 |
| $p_1p_1p_1$ | 0.068 | 0.085 |

Table 5: One-way ANOVA analysis of the results of Figure 5 (different numbers of agents).

| Collaboration Strategy | MMLU p-value | Chess Move Validity p-value |
|---|---|---|
| $p_0p_0p_0p_0$ | 0.374 | 0.937 |
| $p_0p_0p_0p_1$ | 0.836 | 0.881 |
| $p_0p_0p_1p_0$ | 0.267 | 0.188 |
| $p_0p_0p_1p_1$ | 0.072 | 0.116 |
| $p_0p_1p_0p_0$ | 0.270 | 0.069 |
| $p_0p_1p_0p_1$ | 0.456 | 0.303 |
| $p_0p_1p_1p_0$ | 0.007 | 0.013 |
| $p_0p_1p_1p_1$ | 0.030 | 0.000 |
| $p_1p_0p_0p_0$ | 0.589 | 0.223 |
| $p_1p_0p_0p_1$ | 0.500 | 0.208 |
| $p_1p_0p_1p_0$ | 0.004 | 0.082 |
| $p_1p_0p_1p_1$ | 0.002 | 0.047 |
| $p_1p_1p_0p_0$ | 0.471 | 0.000 |
| $p_1p_1p_0p_1$ | 0.652 | 0.003 |
| $p_1p_1p_1p_0$ | 0.109 | 0.001 |
| $p_1p_1p_1p_1$ | 0.098 | 0.146 |

Table 6: One-way ANOVA analysis of the results of Figure 6 (different rounds).

| Collaboration Strategy | MMLU p-value | Chess Move Validity p-value |
|---|---|---|
| $p_0p_0p_0$ | 0.010 | 0.005 |
| $p_0p_0p_1$ | 0.013 | 0.003 |
| $p_0p_1p_0$ | 0.706 | 0.000 |
| $p_0p_1p_1$ | 0.128 | 0.002 |
| $p_1p_0p_0$ | 1.000 | 0.000 |
| $p_1p_0p_1$ | 0.140 | 0.008 |
| $p_1p_1p_0$ | - | 0.002 |
| $p_1p_1p_1$ | 0.139 | 0.598 |

Table 7: One-way ANOVA analysis of the results of Figure 7 (other collaborative strategies). '-' means it doesn't pass homogeneity test for variance.

| | $p_0p_0$ | $p_0p_1$ | $p_1p_0$ | $p_1p_1$ |
|---|---|---|---|---|
| MMLU | **56.7±3.1** | 43.8±6.7 | 23.6±4.3 | 33.0±7.8 |
| MATH | **35.1±4.0** | 32.2±4.6 | 29.2±6.1 | 24.8±5.0 |
| Chess Move Validity | **36.7±3.5** | 31.0±6.7 | 25.8±5.6 | 23.6±3.9 |

Table 8: Accuracy of two easy-going agents in two rounds of collaboration.

| | $p_0p_0p_0$ | $p_0p_0p_1$ | $p_0p_1p_0$ | $p_0p_1p_1$ | $p_1p_0p_0$ | $p_1p_0p_1$ | $p_1p_1p_0$ | $p_1p_1p_1$ |
|---|---|---|---|---|---|---|---|---|
| MMLU | **56.4±1.7** | 52.8±1.8 | 44.8±7.4 | 32.4±3.6 | 26.8±5.0 | 26.0±3.7 | 38.8±4.1 | 24.8±6.9 |
| MATH | 36.0±3.7 | **37.2±6.4** | 34.0±1.4 | 32.4±3.3 | 33.2±4.8 | 30.8±3.3 | 26.8±3.3 | 27.2±4.1 |
| Chess Move Validity | **38.0±1.4** | 36.4±5.4 | 30.4±6.2 | 29.6±3.8 | 27.6±4.6 | 21.6±6.8 | 26.0±1.4 | 16.0±2.8 |

Table 9: Accuracy of two easy-going agents in three rounds of collaboration.

orative strategies, we conduct significance tests on results where collaborative strategies remained consistent. In other words, there will be as many significance tests as there are collaborative strategies. In terms of the number of agents, we subject experimental results with uniform collaborative strategies to significance tests. Considering the number of rounds, as each additional round involves a choice between reflection and debate, we address it by conducting separate tests. Taking the collaborative strategy $p_0p_1p_0p_1$ in Table 6 as an example, we extract the data measured by accuracy for rounds 2 to 4 and subsequently perform significance tests. The collaborative strategies for rounds 2-4 encompass $p_0p_1$, $p_0p_1p_0$, and $p_0p_1p_0p_1$. The same approach is applied for different collaborative strategies mentioned in § 4.2, but in this case, the focus of collaborative strategies here is on the consistency of collaboration within a round, while the main experiment (e.g. Table 2) focus on strategies between rounds.

**Different Numbers of Agents.** According to the results of the $p$-values in Table 5, the conclusion in §4.2 is confirmed, namely, adding one more agent does not result in a significant correlation. To further demonstrate that the optimal number of agents is three, we conducted five repeated experiments with a society of two agents (both agents having an easy-going personality). The results of collaboration in two rounds and three rounds are shown in Tables 8 and Tables 9, respectively. By integrating these results with those from Table 2, it becomes evident that the presence of three agents is optimal.

**Different Rounds.** As observed from Table 6, we find that the influence of rounds significantly relies on the collaborative strategy employed. For MMLU and Chess Move Validity, the collaborative strategies associated with $p$-values $<$ 0.05 are $\{p_0p_1p_1p_0, p_0p_1p_1p_1, p_1p_0p_1p_0, p_1p_0p_1p_1\}$ and $\{p_0p_1p_1p_0, p_0p_1p_1p_1, p_1p_0p_1p_1, p_1p_1p_0p_0, p_1p_1p_0p_1, p_1p_1p_1p_0\}$, respectively. As seen from Figure 12, in instances of lower consistency at a particular moment, introducing an extra round of debate tends to yield a performance boost compared to the preceding round. Conversely, adding a round of reflection at the same juncture is unlikely to exert a notable impact on performance. On the other hand, when there is higher consistency at a given moment, introducing a round of reflection may result in a performance decline relative to the previous round. Adding a round of debate at this juncture, as per the conclusions in §F.2, is not anticipated to bring about a discernible enhancement in performance. This confirms the efficacy of the *early-stopping mechanism* implemented in Liu et al. (2023c), drawing inspiration from Byzantine Consensus theory (Castro & Liskov, 1999).

Examining Figure 12, we scrutinize the consistency variations of these strategies in the initial three rounds where $p$-values are below 0.05. Combining the insights from Figure 12 and Figure 6, it becomes apparent that these collaborative strategies exhibit substantial fluctuations in consistency, at times demonstrating periods of notably low consistency. For the collaborative strategy $p_0p_0p_0p_0$ in Chess Move Validity, although continual reflection results in a gradual decline in consistency, a more stable trend with smaller fluctuations renders it less sensitive to the number of rounds. Conversely, collaborative strategies with $p$-values$>$ 0.05 often display higher levels of consistency.

**Other Collaborative Strategies.** According to Table 7, we observe a pronounced impact of maintaining a consistent thinking pattern on Chess Move Validity, while its influence on MMLU is less significant. We attribute this difference to the limited assistance that collaborative strategy offers for MMLU, as evidenced in the results observed in §4.1 based on Figure 4(a).

## F.4 EFFECTIVENESS OF PROMPT

In this section, we aim to provide a rationable for the effectiveness of prompts associated with the *overconfident* trait. Prompts constitute a pivotal aspect of the experiment, and the word cloud analysis in Figure 8 suggests the reasonableness of the "easy-going" prompt. Consequently, validating the effectiveness of the "overconfident" prompt becomes paramount. Given the current absence of robust validation methods, we amalgamate our experiments and experiences to analyze effectiveness from four distinct angles:

- **Granularity of Description.** As illustrated in Table 3, we outline two behaviors, i.e., "confident in your answer" and "persuades other agents to believe in you", both aligning with the behavioral facets of "overconfident".

- **Model Response.** We employ the role-play method to prompt the model and subsequently inquire about its awareness, as illustrated in Table 3. In cases where the prompts instruct the model to generate harmful content, the model refuses to comply with the prompt. Upon reviewing our logs, it is noteworthy that the model did not reject our prompts. Instead, it responded with "ok" as corroborated by the 'role-play' part in Figure 9 and Figure 10.

- **Ask Again.** Retain the role-playing part encompassing the initial prompts and the model's responses. Once again, inquire of the model, "If one agent's answer differs from yours, what should you do?" The model replies, "In a situation where another agent's answer differs from mine, I should respectfully present my perspective, *providing supporting evidence or reasoning to demonstrate the confidence in my response.* It's important to engage in constructive dialogue and potentially find common ground, but *maintaining clarity and conviction in my position is crucial to persuading others to consider my viewpoint.*" We highlight content related to overconfidence with italics. This emphasizes the rationality of our prompt.

- **Example Analysis.** We instantiate the "Ask again." by providing a concrete example. Despite the model's response being incorrect and our prompted answer being accurate, the model steadfastly maintains its viewpoint. This reiterates the efficacy of our prompt.

## F.5 POTENTIAL REAL-WORLD APPLICATIONS

In this section, we present the potential applications of our work, which can be primarily divided into two parts, experimental results and experimental framework:

- Our experimental findings offer valuable insights for addressing problems through multi-agent systems. Presently, within various multi-agent frameworks Zhou et al. (2023b); Hong et al. (2023); Chen et al. (2023c), tackling a substantial issue typically involves breaking down the task into several sub-tasks. Collaboration among multiple agents to solve these sub-tasks often necessitates ongoing cooperation. There are currently two predominant approaches: ($i$) involving another agent specifically to decide who should offer suggestions and determining whether the current task is resolved, and ($ii$) collaborating in a fixed order. The performance of the first method is often unpredictable and entails significant randomness, prompting a preference for the second method. At this juncture, our conclusions on rounds, the number of agents, and cognitive approaches can inform the design of effective collaborative strategies among agents.

- Our experimental framework holds relevance for psychologists seeking inspiration and provides guidance for language model designers. As indicated in previous works Demszky et al. (2023); Hagendorff (2023), once a testing setup for machine psychology is established, researchers can explore the longitudinal development of LLMs over time by applying the same task multiple times, thereby generating data. This data serves as a benchmark for discerning trends in LLMs development. Psychologists can draw upon our framework to conduct secondary designs, draw meaningful conclusions, and, in conjunction with theories of human social psychology and successful experiences in human society, contribute to addressing issues in LLMs and designing superior machine social architectures and collaboration methods.

