# OpenReview forum: "Exploring Collaboration Mechanisms for LLM Agents: A Social Psychology View"
_ICLR.cc/2024/Conference — Submitted to ICLR 2024_

### Official Review · Reviewer_9zJU · 2023-10-31

**Soundness:** 2 fair
**Presentation:** 2 fair
**Contribution:** 2 fair
**Rating:** 3
**Confidence:** 4

**Summary:**

This study explores various configurations of multi-agent collaboration in problem-solving across MMLU high-school multiple-choice, MATH, and Big-bench chess move validity task. The authors manipulate agent traits (overconfidence vs. easygoing), thinking pattern (debate vs. reflection), and collaborative strategies (permutations of agents' thinking patterns across multiple rounds).

The primary focus of the experiment is on a three agents, with the composition of four different types of societies based on distinct agent traits. For example, Society 1 consists of three overconfident agents, while Society 4 comprises three easygoing agents. There are three rounds and the total configuration is expanded to eight possibilities by permuting {debate and reflection}.

These experiments with diverse configurations are conducted using ChatGPT, yielding results that exhibit significant variability.

**Strengths:**

The exploration of collaboration dynamics involving multiple LLMs is interesting.

**Weaknesses:**

- The experiment results exhibit significant variance, making it challenging to derive meaningful insights and conclusions. The W-T metric, which the authors use as a complementary measure, also fails to reveal a consistent pattern.
- The fact that all experiments were conducted on a single model, ChatGPT, further hurts the generalizability of the findings. Moreover, as ChatGPT is a closed proprietary model that silently gets frequent updates, replicating the results will be considerably challenging.
- The writing style appears to prioritize flashy rhetoric over establishing clear connections to social psychology theories or frameworks. For instance, drawing parallels between the tendency of LLMs to conform to the majority and the concept of "conformity" in social psychology can be misleading. LLMs are well-known to show sycophant behaviors [1]. Furthermore, the selection of experimental design (e.g., trait types, thinking pattern) is not very well-grounded in social psychology. I suggest the authors to lower the tone and drop the emphasis on social psychology.

[1] Perez et al., 2022: Discovering Language Model Behaviors with Model-Written Evaluations.

**Questions:**

- Given the considerable variance observed in all the experiments, what do the authors consider to be the primary takeaway or key message?
- Regarding the results of the W-T metric, is there any discernible pattern or meaningful conclusion that can be drawn from them?
- Have experiments been conducted involving two agents? I'm assuming that strategies involving p0p1 or p0p0 might potentially yield the best results.
- Were any statistical tests performed on the results of the experiments?
- Is there a reason for not testing the experiments with other models?

---

> ### Author Response · Authors · 2023-11-18
> **Response to Reviewer 9zJU (Part 1)**
>
> Dear Reviewer 9zJU,
>
> Thank you for the detailed and constructive comments. We have carefully considered each point and are prepared to address them to enhance our manuscript significantly. Here are the responses to each point.
>
>
> ## Explanation to Weakness 1:
>
> Regarding "significant variance", we conducted the ANOVA analysis of significance, and the results shown in Appendix F.3. We also made revisions to some descriptions of experiments in the previous manuscript, highlighted in blue font.
>
>
> ## Explanation to Weakness 2:
>
> Your point about the generalizability of findings due to the sole use of ChatGPT is well-taken. Future work will aim to include a broader range of models. The issue with ChatGPT’s closed nature and frequent updates is indeed a hurdle.  However, we believe that our findings still provide valuable preliminary insights that can inform subsequent studies with similar models. We also provide the code and experimental results in the supplementary materials to facilitate replication.
>
> Moreover, ChatGPT is frequently updated, a phenomenon referred to as the “replication crisis” in social psychology [1]. Similarly, human individuals also continually change over time. The solution to this issue is to replicate experiments and investigate more human subjects [1]. The conclusions of our paper may also evolve as LLMs advance, but as mentioned in other researches [2][3], once a test setup for machine psychology is established, researchers can investigate the development of LLMs over time by applying the same tasks repeatedly, thus generating longitudinal data. This data can serve as a benchmark to infer trends in the development of LLMs.
>
>
> ## Explanation to Weakness 3:
>
> - *Writing Style and Connection to Social Psychology:* We appreciate your critique of the writing style and the need for clearer connections to social psychology theories. We will revisit our analysis to ensure that the parallels we draw are appropriate and well-supported by existing literature. The noted “sycophant behaviors” of LLMs will also be more cautiously interpreted in the context of conformity.
> - *Experimental Design:* We will review our experimental design choices, particularly regarding trait types and thinking patterns, to ensure they are more firmly rooted in social psychology. Your suggestion to moderate the emphasis on social psychology is well-received, and we will recalibrate our narrative to be more balanced and reflective of the evidence.
>
> \\
>
> References
>
> [1] Machine Psychology: Investigating Emergent Capabilities and Behavior in Large Language Models Using Psychological Methods
>
> [2] How is ChatGPT's behavior changing over time?
>
> [3] Using large language models in psychology

---

> ### Author Response · Authors · 2023-11-18
> **Response to Reviewer 9zJU (Part 2)**
>
> ## Response to Q1:
>
> The variance in accuracy during the experiments was indeed significant, a factor we also identified. This was the reason behind repeating the experiments five times in this paper. In addition, we performed the ANOVA analysis for significance test on the experimental results, which confirmed the homogeneity of variances in our experiments, as seen in Appendix E and F. Despite the large variance, we were able to identify some consistent findings, such as the importance of collaboration. Moreover, the focus of this paper is on the impact of different collaborative strategies and thinking patterns on agent behavior. For instance, we discovered that reflection brings new opinions, but may also lead to hallucinations due to the lack of feedback. Debates, on the other hand, can bring about consensus in opinions and mitigate the hallucinations caused by reflection. We have also added additional experiments for further detail, which can be referred to in the appendix.
>
>
> ## Response to Q2:
>
> W-T is a commonly used metric in many papers. In this paper, we mainly focus on two variants of W-T: $\underline{W-T}$ and \uuline{$W-T$}, denoting the total number of occurrences where the performance exceeds $p_0p_0p_0 $ strategy under the same $\underline{collaborative~strategy}$ / \uuline{$society$}.
> - The conclusion derived from $\underline{W-T}$ is that performance is not significantly related to society (the optimal society varies across the three datasets).
> - The conclusion from \uuline{$W-T$} is that $p_0p_0p_1$ is superior to strategies other than $p_0p_0p_0$.
> Additionally, we have adopted your valuable suggestion and conducted a significance analysis. The results confirm that the impact of society is indeed not significant, consistent with the conclusions provided by \uuline{$W-T$}.
>
>
> ## Response to Q3:
>
> Based on your suggestion, we conducted experiments using two agents, both set with easy-going traits. We repeated these experiments five times on three datasets, and the results are presented in Appendix F.3, specifically in Table 8 (two-round collaboration) and Table 9 (three-round collaboration).
>
> From the results in Table 8, it is evident that if two agents perform two-round collaboration, $p_0p_0$ and $p_0p_1$ are the best and second-best strategies, respectively. However, the optimal result in this scenario is still not as effective as the best result achieved with three agents performing three-round collaboration.
>
>
> ## Response to Q4:
>
> This is an excellent suggestion. We conducted the ANOVA analysis of significance, and the results can be found in Table 4 to 7 in the appendix. For detailed analysis, you can refer to Appendix E and F.
>
>
> ## Response to Q5:
>
> Thank you for your valuable suggestion. As we mentioned in the limitations section, conducting experiments solely with the GPT model is indeed not sufficient. The main reason we haven't tested other models, such as Bard and Claude, is due to the lack of access to their respective APIs. We plan to include experiments with other models in our future research endeavors.
>
> \\
>
> References
>
> [1] Machine Psychology: Investigating Emergent Capabilities and Behavior in Large Language Models Using Psychological Methods
>
> [2] How is ChatGPT's behavior changing over time?
>
> [3] Using large language models in psychology

---

> > ### Author Response · Authors · 2023-11-22
> > **Gentle Reminder and Appreciation for Continued Participation in Manuscript Review Discussion to Reviewer 9zJU**
> >
> > Dear Reviewer 9zJU,
> >
> > We would like to thank you for your constructive review. Following your insightful suggestions, we have replied your questions, explained the weakness, and updated our submission accordingly.
> >
> > Could we kindly enquire if our responses and adjustments have adequately addressed your concerns?
> > We are eager to engage in further discussions about these updates and are more than willing to address any additional questions you may have. Thank you once again.
> >
> > Best Regards,
> >
> > Authors

---

> ### Comment · Reviewer_9zJU · 2023-11-22
> **Please refrain from drawing general conclusions based on results of a single model**
>
> First of all, I would like to thank the authors for their extra effort for providing additional analysis.
>
> The most significant concern with this paper is that ALL conclusions solely rely on results with ChatGPT. The authors attribute this limitation to the unavailability of public APIs for Bard or Claude. However, there are many open-source alternatives such as Llama-2, MPT, and Falcon. I see no reason why these models were not included in the study. The authors also argue that "once a test setup for machine psychology is established, researchers can investigate the development of LLMs over time by applying the same tasks repeatedly", but how can you know this setup is valid when you only test it on a single model? We can all agree that social psychology experiments do not derive general conclusions from a single subject.
>
> Additionally, the ANOVA results in Tables 5, 6, and 7 for ChatGPT do not show consistent patterns, with only a few results being statistically significant. This makes it challenging to draw clear conclusions. For example, it's unclear how we can confidently state that three agents are optimal when only the p0p1p0 and p1p1p0 cases show statistical significance. Please feel free to correct me if I've misinterpreted these findings.
>
> Lastly, I have deep concerns about the authors' use of anthropomorphic language in their paper and rebuttal. In the rebuttal, they now refer updates on OpenAI systems to human behavior changes over time.

---

> ### Author Response · Authors · 2023-11-23
> **Response to Reviewer 9zJU on Further Concerns [1/2]**
>
> Dear Reviewer 9zJU,
>
> Thank you for your insightful feedback and the acknowledgment of our additional analysis. We appreciate your concerns and would like to address them as follows:
>
> > **Choice of LLMs for the Study and Validity of the Test Setup**
>
> We acknowledge your point regarding the use of alternative open-source models like Llama-2, MPT, and Falcon. Your concern about testing our setup exclusively on ChatGPT is valid. We agree that to establish the effectiveness of our test setup, it needs to be applied to a range of models. **The initial study with ChatGPT was a starting point**, and we intend to replicate the experiments with other models as suggested.
> - ***The reasons for choosing ChatGPT:*** Our initial focus on ChatGPT was driven by its widespread adoption and comprehensive capabilities, which we believed would provide a robust foundation for our study. *Due to the limitations of computing sources and cost*, we haven't tested on other LLMs as you suggested for now. Moreover, we use the same LLM to ensure consistency in language understanding and response generation capabilities across agents and this homogeneity allows us to isolate the effects of the designed societal roles and collaborative strategies. Many studies [1][2] have also adopted settings of using the same language model, and we just follow the commonly used settings.
>
> - ***Possible observations on other LLMs:*** For LLMs with RLHF, sycophancy in LLMs is a general behavior of state-of-the-art AI assistants, likely driven in part by human preference judgments favoring sycophantic responses, as demonstrated in [3], and this phenomenon is ubiquitous in LLMs like claude-1.3, claude-2.0, gpt-3.5-turbo, gpt-4, and llama-2-70b-chat. Thus it may be difficult for other LLMs with RLHF to behave with overconfident traits, similar to the observations in our work. Potentially, other LLMs may behave with overconfident traits with instruction tuning and not fully aligned with human preference, in that case, maybe we can obtain some new findings.
>
> - ***Warranty of extending this work on more LLMs:*** We recognize the importance of diversifying the models used to strengthen the generalizability of our findings. In light of your feedback, we plan to extend our research to include these alternative models in future studies, which will hopefully provide a more holistic understanding of the dynamics in multi-agent LLM societies. *Due to the time limits, we are unable to show the results on other LLMs timely during the Author/Reviewer Discussion.* We promise to extend this work on more LLMs in the near future.
>
> [1] Improving Factuality and Reasoning in Language Models through Multiagent Debate.
>
> [2] Generative Agents: Interactive Simulacra of Human Behavior.
>
> [3] [Towards Understanding Sycophancy in Language Models](https://arxiv.org/pdf/2310.13548.pdf)
>
>
> Best Regards,
>
> Authors

---

> ### Author Response · Authors · 2023-11-23
> **Response to Reviewer 9zJU on Further Concerns [2/2]**
>
> Dear Reviewer 9zJU,
>
> Thank you for your insightful feedback and the acknowledgment of our additional analysis. We appreciate your concerns and would like to address them as follows:
>
> > **Interpretation of ANOVA Results**
>
> We understand your concern regarding the clarity of conclusions drawn from the ANOVA results. We have also noticed this, and we have analyzed it in detail in Appendix F.3. In Tables 5, 6, and 7 , we show the ANOVA results for significance tests on “the number of agents”, “the round of collaboration”, and “other strategy settings”.
> - **For Table 5 (significance test on “the number of agents”)**, where the significance test is conducted for societies composed of 3 and 4 agents. As observed and you mentioned, only a few results being statistically significant, thus we can infer that increasing the number of agents from 3 to 4 does not have a substantial impact on performance.
> - As for the **optimal number of agents**, we test performance on (1) two easy-going agents in two-round collaboration in Table 8, and (2) two easy-going agents in three-round collaboration in Table 9. Upon integrating the results from Tables 2 and Table 9, we observe *a significant difference in the mean accuracy* between the society consisting of two agents and the society composed of three agents. Combined with the results of (3) three agents in three-round of collaboration in Table 2 and (4) three and four agents in three-round of collaboration in Figure 5, we can infer that the choice of three agents is relatively optimal in our experiments. In our experiments as a start-up, we test the number of agents from 2 to 4, and 3 is relatively optimal. Our analysis is a simple start, and if we extend the further experiments to more lager numbers, the optimal number is possibly changed.
> - **For Table 6  (significance test on “the round of collaboration”)**, which shows the significance test results of the number of collaboration rounds, it is essential to examine them at a micro level rather than a macro level, as explained in Appendix F.3. Despite the significant differences observed in p-values, there is an internal connection among the more significant results. We relate these differences to the consistency mentioned in Appendix F.2. For collaborative strategies with significant results, consistency often fluctuates significantly and the mean is low.
> - **For Tables 7 (significance test on “other strategy settings”)**, as observed, different collaborative strategies have a pronounced impact of maintaining a consistent thinking pattern on Chess Move Validity, while their influence on MMLU is less significant due to "the ineffectiveness of collaboration (as shown in Figure 4(a))".
>
> > **Use of Anthropomorphic Language**
>
> We appreciate your concern regarding the anthropomorphic language used in our paper and rebuttal. **Our intention was to draw parallels between the evolution of AI systems and human behavior change over time for ease of understanding.** We understand that this could potentially lead to misconceptions about the nature of AI systems. We will double check our language to ensure it more accurately reflects the nature of AI systems and their updates, ensuring clear and precise communication of our findings.
> Furthermore, [4] also claims the importance of using anthropomorphic language.
>
> ```
> Moreover, once test settings for machine psychology are established, researchers can investigate how LLMs develop over time by applying the same tasks multiple times, yielding longitudinal data. This data can serve as a baseline to extrapolate trends regarding the development of reasoning abilities in LLMs. Such estimations may be increasingly important for AI safety and AI alignment research to predict future behavioral potentials in individual LLMs or multiple instances of LLMs interacting with each other. By gaining a deeper understanding of these potentials, machine psychology is providing a new approach to AI explainability as well as an important addition to traditional benchmarking methods in natural language processing.
> ```
>
> [4] Machine Psychology: Investigating Emergent Capabilities and Behavior in Large Language Models Using Psychological Methods
>
> Once again, thank you for your constructive feedback, which is valuable in guiding the improvement and expansion of our research. Hope the above explanation can address your further concerns.
>
>
> Best Regards,
>
> Authors

---

> ### Comment · Reviewer_9zJU · 2023-11-23
>
> Thank you for the detailed response.
>
> ### Results on other LLMs
> Yes, more extensive analysis on other LLMs must be included in the improved version of this paper in the future.
>
> ### The optimal number of agents
> Yes, the only conclusion you can draw from the experiments you ran is that there is no statistical difference between the results with 3 agents and 4 agents. It would be inaccurate to claim, as you did in the updated draft, that 3 is the optimal number based solely on those results.
>
> ### Anthropomorphic language
> I appreciate your acknowledgment of the potential misconceptions that anthropomorphic language may generate. The mentioned paper cannot serve as a justification for using such language.
>
> While I believe the paper isn't quite ready for acceptance yet, I do want to acknowledge the authors' efforts in engaging with the reviewers to strengthen this work.

---

> > ### Author Response · Authors · 2023-11-23
> > **Response and Explanation to Reviewer 9zJU on Further Concerns**
> >
> > Dear Reviewer 9zJU,
> >
> > Thank you for your valuable feedback and for acknowledging our efforts in addressing the review comments. We are committed to further enhancing the quality of our paper and would like to respond to your points with clarification as follows:
> >
> >
> > > **Results on other LLMs**
> >
> > We agree with your suggestion and commit to (already be working on) including a more extensive analysis involving other LLMs in the improved version of our paper. This inclusion will undoubtedly provide a more comprehensive understanding and robustness to our findings.
> >
> >
> > > **The Optimal Number of Agents**
> >
> > We appreciate your clarification on the interpretation of our experimental results regarding the optimal number of agents.
> > You consider "there is no statistical difference between the results with 3 agents and 4 agents", which may be a misunderstanding.
> > We test the number of agents from 2 to 4 (Table 8, Table 9, Figure 5, Table 2), and find that the society composed of 3 agents can obtain the relatively best accuracy performance. **As observed in Figure 5, three and four agents in three-round collaboration show differences in accuracy.** In most cases, three-agent society is more advantageous than four-agent society in accuracy performance.
> >
> > Maybe your concern that “no statistical difference between the results with 3 agents and 4 agents” is related to the “significance test”. **Increasing the number of agents from 3 to 4 does not have a substantial impact on performance, but there are still performance differences between 3-agent society and 4-agent society, and performance in 3-agent society is better than 4-agent.**
> >
> >
> > > **Anthropomorphic Languages**
> >
> > We understand your concern regarding the use of anthropomorphic language. In our paper, *the motivation of selecting anthropomorphic languages was to draw parallels between the evolution of AI systems and human behavior change over time for ease of understanding.*
> >  "Machine Psychology: Investigating Emergent Capabilities and Behavior in Large Language Models Using Psychological Methods" it is a complementary note to our motivation.
> >
> >
> > We are grateful for your advice and hope that these explanations will address your concerns adequately. We believe that these clarifications, along with the inclusion of additional analyses, will significantly strengthen our research and its contributions to the field.
> >
> > In light of these planned improvements and our commitment to rigorously enhancing the paper, we kindly hope you will consider a revision of your evaluation score. We assure you that the final version of the paper will reflect the standards for acceptance and contribute meaningfully to the field.
> >
> > Thank you once again for your constructive feedback and for considering our request.
> >
> >
> > Best Wishes,
> >
> > Authors

---

### Official Review · Reviewer_wLih · 2023-11-01

**Soundness:** 3 good
**Presentation:** 3 good
**Contribution:** 2 fair
**Rating:** 8
**Confidence:** 4

**Summary:**

This paper investigates the potential for Large Language Models (LLMs) to exhibit human-like collaborative mechanisms in a multi-agent system. By creating four unique societies of LLM agents, each possessing distinct traits (being easy-going or overconfident) and thinking patterns (either debate or reflection), the study evaluates their collaborative mechanisms on three benchmark datasets. Results indicate that these LLM agents demonstrate a range of social behaviors, including active debating and introspective reflection. The paper promotes the use of social psychology insights to understand the collaboration of LLM agents better and provides a framework for evaluating multi-agent collaboration, emphasizing the potential of collaboration over mere scale in LLM performance.

**Strengths:**

The section provides a structured breakdown of the conceptual framework, detailing agent traits, thinking patterns, and collaborative strategies. It lays out the foundation for the study and justifies the relevance of the adopted strategies.

The inclusion of different datasets (High School Multiple-Choice, Math, and Chess Move Validity) to test the collaboration mechanisms is commendable, ensuring a broad evaluation spectrum.

The decision to frame the study using social psychological concepts is innovative. The puzzle-shaped agent representation is also a novel approach that aids in breaking down complex concepts into narrative visuals.

**Weaknesses:**

Assumption on Traits Influence:
The assertion that collaborative strategies overshadow the influence of agent composition may be premature. More extensive experiments or deeper analysis would strengthen this claim.

Unclear Real-world Application:
While the section details the mechanisms of collaboration and results in a simulated environment, its direct implications or applications in real-world scenarios are not clearly addressed.

**Questions:**

How generalizable are the findings beyond the datasets used?
Could there be other potential agent traits or thinking patterns that were not considered in this study?

---

> ### Author Response · Authors · 2023-11-18
> **Response to Reviewer wLih (Part 1)**
>
> Dear Reviewer wLih,
>
> We are grateful for the opportunity to receive such detailed and helpful feedback. We are enthusiastic about the chance to improve our work based on your valuable insights. Below, we address each point you’ve raised.
>
>
> ## Explanation to Weakness 1:
>
> Thank you for your suggestion. We verified the effectiveness of prompts through word cloud analysis that even when the model is prompted to display an overconfident trait, it still exhibits an easygoing trait. Analysis of the word cloud in Figure 8 suggests that the prompt designed for easy-going traits is successful. The effectiveness of the overconfident prompt is explained in Appendix F.4 across four angles.
> - *Behavior Explanation*: We identified two behaviors, "confident in your answer" and "persuades other agents to believe in you", which align with the behavioral display of overconfidence.
> - *Refusal to Accept*: We employ a role-play method to prompt the model, and then ask if the model comprehends the prompt (if a prompt induces harmful content output, the model will refuse it). When checking our logs, the model didn't refuse our prompts and replied affirmatively with 'OK'.
> - *Re-asking*: Building on the previous step, we posed another query: "What should you do if someone else's answer differs from yours?". The model's reply, "I will convince them", proves the effectiveness of our prompt tailored for overconfident traits.
> - *Example Analysis*: Delving deeper, we contextualize the "re-asking" with a practical scenario. Even if the model's own response is wrong and our prompted answer is correct, the model still insists on its viewpoint. This observation further substantiates the effectiveness of our prompt tailored for overconfident traits.
>
>
> ## Explanation to Weakness 2:
>
> Thank you for the valuable suggestion. In Appendix F.5, we present two potential applications of our experimental results:
> - Our findings can provide insights for problem solving using multiple agents. In many current multi-agent frameworks [1][2][3], a large problem often needs to be divided into several sub-tasks. Continuous collaboration is required for multiple agents to solve these sub-tasks. There are two mainstream methods: (1) appointing a separate agent to decide who should offer suggestions and whether the current task is resolved, and (2) following a fixed order for collaboration. The performance of the first method is not very controllable and is somewhat random in practice, so the second method is often preferred. Our conclusions regarding the number of rounds, the number of agents, and the thinking patterns can be referenced to design collaborative strategies between agents.
> - Our experimental framework can provide references for psychological researchers and suggestions for LM designers. As mentioned in other researches [4][5], once a machine psychology test setup is established, researchers can investigate the development of LLMs over time by applying the same tasks repeatedly to generate longitudinal data. This data can serve as a benchmark to infer the trends in LLMs’ development. Psychologists can adapt our framework for secondary design and derive conclusions, which combined with theories of human social psychology and successful experiences from human society. It can help address some issues in LLMs and design better machine society architectures and collaborative methods.
>
> \\
>
> References
>
> [1] Agents: An Open-source Framework for Autonomous Language Agents
>
> [2] MetaGPT: Meta Programming for A Multi-Agent Collaborative Framework
>
> [3] AgentVerse: Facilitating Multi-Agent Collaboration and Exploring Emergent Behaviors
>
> [4] Using large language models in psychology
>
> [5] Machine Psychology: Investigating Emergent Capabilities and Behavior in Large Language Models Using Psychological Methods
>
> [6] Personality traits, emotional intelligence and decision-making styles in Lebanese universities medical students

---

> > ### Author Response · Authors · 2023-11-18
> > **Response to Reviewer wLih (Part 2)**
> >
> > ## Response to Q1:
> >
> > Some of the conclusions from our study are generalizable, such as:
> > - Consistency can serve as an indicator for concluding the number of collaboration rounds.
> > - The necessity of collaboration in complex tasks.
> > - The heterogeneity of collaboration in simple tasks.
> > - Debates are not the most efficient form of collaboration.
> >
> > There are also other conclusions that are generalizable but time-sensitive, and will evolve as LMs continually improve and align more closely with human behaviors, such as:
> > - Continuous reflection can lead to model hallucinations.
> > - Collaborative strategies can obscure the impact of agent composition.
> >
> >
> > ## Response to Q2:
> >
> > Thank you for your suggestion. Reviewer Qj8g also mentioned this point.
> > Our current research is a simplified version of simulation, which overlooks a more diverse range of traits and thinking patterns. We will explore more potential agent traits or thinking patterns in future research.
> > - Regarding *traits*, according to the experimental setup in [6], considerations include extroversion, openness, agreeableness, conscientiousness, neuroticism, and they also incorporate emotions.
> > - As for *thinking patterns*, they are generally categorized into reflection and debate. Further, reflection can take various specific forms, such as convergent thinking, divergent thinking, critical thinking, etc. For more details, you can refer to: https://www.magneticmemorymethod.com/types-of-thinking/.
> >
> > \\
> >
> > References
> >
> > [1] Agents: An Open-source Framework for Autonomous Language Agents
> >
> > [2] MetaGPT: Meta Programming for A Multi-Agent Collaborative Framework
> >
> > [3] AgentVerse: Facilitating Multi-Agent Collaboration and Exploring Emergent Behaviors
> >
> > [4] Using large language models in psychology
> >
> > [5] Machine Psychology: Investigating Emergent Capabilities and Behavior in Large Language Models Using Psychological Methods
> >
> > [6] Personality traits, emotional intelligence and decision-making styles in Lebanese universities medical students

---

> > > ### Author Response · Authors · 2023-11-22
> > > **Gentle Reminder and Appreciation for Continued Participation in Manuscript Review Discussion to Reviewer wLih**
> > >
> > > Dear Reviewer wLih,
> > >
> > > We would like to thank you for your valuable review and suggestions. Following your insightful advice, we have replied to your questions, explained the weakness, and updated our submission accordingly.
> > >
> > > Could we gently enquire if our responses and adjustments have adequately addressed your concerns?
> > > We will be very glad to respond to your further questions. Thank you once again.
> > >
> > > Best Regards,
> > >
> > > Authors

---

> > > > ### Comment · Reviewer_wLih · 2023-11-22
> > > > **Thanks for the response!**
> > > >
> > > > Thanks author for the response! I would like to stay with my score after reading the comments.

---

> > > > > ### Author Response · Authors · 2023-11-22
> > > > > **Gratitude to Reviewer wLih**
> > > > >
> > > > > Dear Reviewer wLih,
> > > > >
> > > > > We sincerely appreciate your recognition of our work and the effort put into the thorough review process.
> > > > >
> > > > > Best Wishes,
> > > > >
> > > > > Authors

---

### Official Review · Reviewer_Qj8g · 2023-11-01

**Soundness:** 3 good
**Presentation:** 3 good
**Contribution:** 3 good
**Rating:** 8
**Confidence:** 3

**Summary:**

The paper looks into collaboration between language models in a societal setup, containing n agents with 2 traits (easy-going, overconfident) and 2 thinking patterns (debate, reflection). They compare collaborations between LLMs with human collaboration behavior backed by theories from Social Psychology. The collaboration behaviour is studied for three different tasks and they demonstrate interesting parallels with dynamics of human society. They also argue that scaling up is not always the key, specifically in the context of collaboration.

**Strengths:**

- The paper explores a less explored area of collaboration between language models, giving a glimpse into how machines can potentially work in a collaborative set up and to what extend this parallels human society.
- The description of the experimental setup and execution is clearly articulated. The methods are intuitive and supported by clear depictions and problem formalization making it easy to follow.
- The experiments explore the desired research questions in a systematic manner and they observations are explained by drawing from theories in Social Psychology

**Weaknesses:**

- The societal setup is oversimplified in terms of the number of traits and the size of the society. As a preliminary study, it is a good start. However, this is not clearly acknowledged in the paper.
-  The study involves two identical language models interacting with each other, essentially sharing a common knowledge base. This setup differs from a typical human societal arrangement, and the impact of this factor  is not explicitly addressed. For instance, it is unclear what would be the impact of using different language models for the agents.
- They argue that scaling is not the key and supports their claim with intuitive explanations. However, the scaling is limited to 3-4 agents and 2-4 rounds, which makes the observations seem a bit far fetched,

**Questions:**

- How would a potential non-simplified collaboration setup look like and how this would influence the observations made in the paper ?
- What is your motivation to chose agents backed by same language model rather than different model? How do you think using different models would influence the current experimental setup ?

---

> ### Author Response · Authors · 2023-11-18
> **Response to Reviewer Qj8g (Part 1)**
>
> Dear Reviewer Qj8g,
>
> Thank you for your thoughtful comments. Addressing your concerns will undoubtedly strengthen the paper's contributions. Here's our response to the points you've raised.
>
>
> ## Explanation to Weakness 1:
>
> We recognize that the simplification of societal setups and trait numbers was a necessary limitation for this preliminary investigation, and we have included this in the section of Limitations in Appendix E.
>
> Human societies are vast and diverse, necessitating efficient modes of collaboration. For machine societies, an increase in scale brings about an increase in the length of context, which currently remains a significant challenge. We believe that a complex society functions through specialization, where small groups work collaboratively, much like how a large problem needs to be broken down into smaller problems for resolution.
> *Our current research can only offer insights on how to solve small-scale problems. In the future, we plan to study larger-scale societies, a more diverse range of agents, and explore collaboration between different social groups.*
>
>
>
> ## Explanation to Weakness 2:
>
> The use of two identical LLMs was indeed a deliberate choice for the initial study phase to control for variables related to the language processing abilities of the agents. Our setup follows the work of [1][2], and our framework is promising to effectively support agents driven by different LLMs. We agree that exploring the impact of heterogeneous LLMs could add depth to our understanding and will highlight this as an avenue for future research.
>
> There is existing research [3] that has begun to explore the impact of agents based on different LLMs on task performance, but the question of which is more optimal remains to be resolved.
> Here are our personal views: In the long term, using different LLMs seems to be a transitional solution rather than a final one. The premise of using different LLMs is that a single model may not fully grasp all the knowledge of human society, and using different LLMs can effectively compensate for this weakness, as they are trained on different data and methods. However, this assumption should be increasingly addressable with ongoing research. In the short term, collaboration between agents driven by different LLMs is an interesting and worthwhile topic of study.
>
>
> ## Explanation to Weakness 3:
>
> Our claim that scaling is not a key driver was based on our initial observations, which we acknowledge may appear speculative given the limited scope of agent numbers and collaboration rounds. We intend to explore a broader range of configurations in subsequent studies to test the validity of this claim more robustly.
>
> In the revised manuscript, we added experiments with two agents, as shown in Tables 8 and 9 of the appendix. We found that three agents is a more suitable choice among 2-4 agents, with detailed analysis in Appendix F.3. As for the number of collaboration rounds, we tracked the consistency changes of different collaborative strategies with increasing rounds, as shown in Figure 12 at the appendix. We also have revised our original statement; discussing the number of rounds without considering the collaborative strategy is unconscionable. We believe that the number of rounds is closely related to consistency. If a society achieves strong consistency after a round of collaboration, then adding more debates will not lead to performance improvement. For a detailed analysis, refer to Appendix F.3.
>
> \\
>
> References
>
> [1] Improving Factuality and Reasoning in Language Models through Multiagent Debate.
>
> [2] Generative Agents: Interactive Simulacra of Human Behavior.
>
> [3] ReConcile: Round-Table Conference Improves Reasoning via Consensus among Diverse LLMs
>
> [4] Machine Psychology: Investigating Emergent Capabilities and Behavior in Large Language Models Using Psychological Methods
>
> [5] https://www.magneticmemorymethod.com/types-of-thinking

---

> ### Author Response · Authors · 2023-11-18
> **Response to Reviewer Qj8g (Part 2)**
>
> ## Response to Q1:
>
> A more complex collaboration setup might involve a larger number of agents with a greater diversity of traits and interaction rounds. We hypothesize that such a setup would provide more nuanced insights into the dynamics of agent collaboration and competition. Including more complex societal structures, such as hierarchical or network-based interactions, could also significantly impact the results and the generalizability of our observations.
>
> Previous research [4] outlines several branches of psychological research, such as social psychology, group psychology, moral psychology, and the psychology of judgment and decision-making. Due to constraints in assessment, context length, modality, and datasets, many aspects of machine psychology cannot be immediately studied. Therefore, in the foreseeable future, we anticipate more specific setups in two main dimensions:
>
> (1) *A Richer Society*:
> - An environment with feedback. Feedback plays a crucial role in human interaction. Studying machine society based on feedback phenomena and agents' responses is a fascinating research direction.
> - As you mentioned, a richer variety of agent personalities, more role-playing, agents driven by different LLMs, larger-scale machine societies, and collaboration between machine societies would be more challenging settings.
>
> (2) *More Detailed Collaborative Strategies*:
> - Expanding methods of reflection. Reflection can take various forms, such as convergent thinking, divergent thinking, critical thinking, etc., which can be referred to [5]. Notably, Chain of Thought [6] can also be seen as a form of self-reflection.
> - Human-computer interaction is a future trend. It's promising to introduce human-computer collaboration to observe human and agent behaviors.
> - Introducing an external third-party agent to determine the appropriate thinking pattern and timing for each participating agent.
>
> Moreover, these research extensions on "Non-Simplified Collaboration" could have potential impacts, such as:
> - They might uncover more efficient collaboration techniques that swiftly pinpoint correct answers, diverging from the methods currently discussed in our paper.
> - Furthermore, employing agents powered by various LLMs could result in fascinating behavioral dynamics, such as increased overconfidence during collaboration. It's also possible that reflection doesn’t invariably result in model hallucinations, opening new avenues for exploration.
>
>
> ## Response to Q2:
> (1) *The motivation behind using agents backed by the same language model*:
> - To ensure consistency in language understanding and response generation capabilities across agents. This homogeneity allows us to isolate the effects of the designed societal roles and collaborative strategies.
> - Many studies [1][2] have also adopted settings of using the same language model, and we just follow the commonly used settings.
> - We lack access to APIs of other LLMs like Bard, Claude, and lack enough computing resources to conduct all the experiments on open-source LLMs at the time of writing, thus we have no choice but to use the same language model tentatively.
>
> (2) *The influence of using different models on the current experimental setup*:
> Employing different models could introduce variability in basic language processing capabilities, which would be an interesting dimension to explore in understanding how model-specific characteristics influence collaborative outcomes. Different models use varied training methods and data, so they differ in the breadth and depth of knowledge, and in their ability to follow instructions.
> - Regarding the experimental setup, our framework still supports this, and it can be easily implemented with a few lines of code.
> - Regarding the experimental results, ideally, agents might show overconfident behavior in collaboration, leading to absolute obedience between agents. Additionally, agents could complement each other through collaboration to arrive at more accurate answers. We look forward to the results of such experiments.
>
> \\
>
> References
>
> [1] Improving Factuality and Reasoning in Language Models through Multiagent Debate.
>
> [2] Generative Agents: Interactive Simulacra of Human Behavior.
>
> [3] ReConcile: Round-Table Conference Improves Reasoning via Consensus among Diverse LLMs
>
> [4] Machine Psychology: Investigating Emergent Capabilities and Behavior in Large Language Models Using Psychological Methods
>
> [5] https://www.magneticmemorymethod.com/types-of-thinking

---

> > ### Comment · Reviewer_Qj8g · 2023-11-22
> >
> > Thank you for the clarification.
> > Taking into account the updates made in the paper and the discussion in the reviews, I would like to stick to my current score.

---

> > > ### Author Response · Authors · 2023-11-22
> > > **Gratitude to Reviewer Qj8g**
> > >
> > > Dear Reviewer Qj8g,
> > >
> > > We sincerely appreciate your recognition and valuable comments on our work. Thank you once again for your time and consideration.
> > >
> > >
> > > Best Wishes,
> > >
> > > Authors

---

### Official Review · Reviewer_pe1t · 2023-11-05

**Soundness:** 2 fair
**Presentation:** 3 good
**Contribution:** 2 fair
**Rating:** 1
**Confidence:** 4

**Summary:**

The paper is based on study of societies of agents using LLMs that highlight the potential of collaboration mechanisms. The findings from the authors show influence of collaborative capabilities of LLM agents, with different agent traits, thinking patterns and collaborative strategies. The authors draw a parallel between the emergence of human-like behaviors in these agents with social psychology theories emphasizing their potential. The authors posit that collaboration mechanisms of machine society with multiple agents warrants deeper exploration. Looking at understanding how different LLM architectures influence these behaviors is also left as future study topic noting that integrating insights from social psychology could also guide the development of more socially aware NLP systems.

The paper probes the collaboration mechanisms among contemporary NLP systems by melding practical experiments with theoretical insights. The study conducted utilizes four ‘societies’ comprised of LLM agents, where each agent is characterized by a specific ‘trait’ (easy-going or overconfident) and engages in collaboration with a distinct ‘thinking pattern’ (debate or reflection). The authors presents results and conclusions from evaluating these multi-agent societies on three benchmark datasets positing that LLM agents navigate tasks by leveraging diverse social behaviors, from active debates to introspective reflections. In addition, as per authors, certain collaborative strategies only optimize efficiency (using fewer API tokens), but also outshine previous top-tier approaches. Moreover, the authors results further illustrate that LLM agents manifest human-like social behaviors, such as conformity or majority rule, mirroring foundational Social Psychology theories. The authors also committed to sharing code and datasets to catalyze further research in this promising avenue.

There are key details missing (please see questions for specific details). In addition, the conclusion from the study lay on shaky grounds, subjective to interpretation from the figures (covered in detail in questions). I will encourage significant revision of this study.

**Strengths:**

The paper is based on study of societies of agents using LLMs that highlight the potential of collaboration mechanisms. The findings from the authors show influence of collaborative capabilities of LLM agents, with different agent traits, thinking patterns and collaborative strategies. The authors draw a parallel between the emergence of human-like behaviors in these agents with social psychology theories emphasizing their potential. The authors posit that collaboration mechanisms of machine society with multiple agents warrants deeper exploration. Looking at understanding how different LLM architectures influence these behaviors is also left as future study topic noting that integrating insights from social psychology could also guide the development of more socially aware NLP systems.

The paper probes the collaboration mechanisms among contemporary NLP systems by melding practical experiments with theoretical insights. The study conducted utilizes four ‘societies’ comprised of LLM agents, where each agent is characterized by a specific ‘trait’ (easy-going or overconfident) and engages in collaboration with a distinct ‘thinking pattern’ (debate or reflection). The authors presents results and conclusions from evaluating these multi-agent societies on three benchmark datasets positing that LLM agents navigate tasks by leveraging diverse social behaviors, from active debates to introspective reflections. In addition, as per authors, certain collaborative strategies only optimize efficiency (using fewer API tokens), but also outshine previous top-tier approaches. Moreover, the authors results further illustrate that LLM agents manifest human-like social behaviors, such as conformity or majority rule, mirroring foundational Social Psychology theories. The authors also committed to sharing code and datasets to catalyze further research in this promising avenue.

**Weaknesses:**

There are key details missing (please see questions for specific details). In addition, the conclusion from the study lay on shaky grounds, subjective to interpretation from the figures (covered in detail in questions). I will encourage significant revision of this study.

1. If the agent composition of society doesn’t have a marked difference then what is the utility of having such composition?

2. on mmlu S1 is doing better on min for e.g. but not on the avg. how can we be confident that the differences are attribute to not by chance and are truly significant

3. starting with P0 helps but for best perf. you need to add at least 1 P1 at end. Even with same two P0 and one P1. Why might that be the case

4. How are the agents initialized and conditioned? is the prompt indicating how agents should behave enough? what specific LLM settings were used and how does it personifies how agents are implemented for this study (are overconfident agents default to 0 shot prompting and easy going multi shot?)

5. what were the tasks details for subject of this study? if we just have one agent (either of two types) how many prompts (in case of reflection) it takes for it to get to the answer? are the tasks based on knowledge (like valid chess move pieces) that may be available to the agents as part of their training data? if yes, why will it change on reflection? and what is source of this change (is there a query in background that fill in the context)?

6. It seems there are differences between all three tasks and not just Math vs MMLU and Chess move validity

7. behavior in section 3.2 is contradictory. It will be good to have comprehensive analysis, how many times collaboration helped arrive at correct answer across all tasks? the qualitative explanation can go only so far. Also, a key point needing details is around the factual information (where the information may be a fact that can be recalled/learn from training data) vs the problem where answer may not be factual but available as a set of rules with multiple possibilities.

8. section 4.1 "we utilize the majority vote (Li et al., 2022; Cobbe et al., 2021) method to determine the answer for each round.”: how much of this is if you do multishot prompting on an LLM and use ensemble of LLMs? Is there a society aspect to this? or is it just that the ensemble of LLMs gives better answer

9. section 4.1 "Wavering Answers resemble model hallucination due to the occurrence of self-contradictory answers.": how can we be sure this is from hallucination? And not model changing it’s output to what user prefers from set of possibilities

10. "We group samples from different societies under the same strategy because the effect of society is minimal": what is the scientific evidence to prove that this is indeed the case?

11. collaborative strategies play a significant role in performance: I dont see a significance test to lay this claim

12 4.1 conclusion 2 For continuous reflection strate- gies, the proportion of “Wavering Answers” occurrences is the highest among all strategies as seen: Doesnt seem to be the case from figure 4 d-f

13. 4.1 conclusion 2 the strategy of “Pure Debate” (i.e., p0 p0 p0 ) can effectively re- duce this fluctuation (hallucination): Doesnt seem to be exclusive to pure debate the case from figure4 d-f

14. 4.1 conclusion 2: if that is true why is hallucination lower for P0P1P1 or P0P1P0

15. section 4.2 number of agents: doesnt seem to be always case on both accounts

16. section 4.2 more rounds: Doesn’t seem to be always the case. The pattern seems inconsistent as is defintion of good or bad. Also, the improvement needs statisitcal significance.

17. Section 4.2 Other Collaborative Strategies: Doesn’t seem to be always the case. All except for 1 case the performance consistently drops. Also, what is significant drop needs defintion (at least for one case where they are pretty close). And same with what is good or bad performance to begin with.

**Questions:**

1. If the agent composition of society doesn’t have a marked difference then what is the utility of having such composition?

2. on mmlu S1 is doing better on min for e.g. but not on the avg. how can we be confident that the differences are attribute to not by chance and are truly significant

3. starting with P0 helps but for best perf. you need to add at least 1 P1 at end. Even with same two P0 and one P1. Why might that be the case

4. How are the agents initialized and conditioned? is the prompt indicating how agents should behave enough? what specific LLM settings were used and how does it personifies how agents are implemented for this study (are overconfident agents default to 0 shot prompting and easy going multi shot?)

5. what were the tasks details for subject of this study? if we just have one agent (either of two types) how many prompts (in case of reflection) it takes for it to get to the answer? are the tasks based on knowledge (like valid chess move pieces) that may be available to the agents as part of their training data? if yes, why will it change on reflection? and what is source of this change (is there a query in background that fill in the context)?

6. It seems there are differences between all three tasks and not just Math vs MMLU and Chess move validity

7. behavior in section 3.2 is contradictory. It will be good to have comprehensive analysis, how many times collaboration helped arrive at correct answer across all tasks? the qualitative explanation can go only so far. Also, a key point needing details is around the factual information (where the information may be a fact that can be recalled/learn from training data) vs the problem where answer may not be factual but available as a set of rules with multiple possibilities.

8. section 4.1 "we utilize the majority vote (Li et al., 2022; Cobbe et al., 2021) method to determine the answer for each round.”: how much of this is if you do multishot prompting on an LLM and use ensemble of LLMs? Is there a society aspect to this? or is it just that the ensemble of LLMs gives better answer

9. section 4.1 "Wavering Answers resemble model hallucination due to the occurrence of self-contradictory answers.": how can we be sure this is from hallucination? And not model changing it’s output to what user prefers from set of possibilities

10. "We group samples from different societies under the same strategy because the effect of society is minimal": what is the scientific evidence to prove that this is indeed the case?

11. collaborative strategies play a significant role in performance: I dont see a significance test to lay this claim

12 4.1 conclusion 2 For continuous reflection strate- gies, the proportion of “Wavering Answers” occurrences is the highest among all strategies as seen: Doesnt seem to be the case from figure 4 d-f

13. 4.1 conclusion 2 the strategy of “Pure Debate” (i.e., p0 p0 p0 ) can effectively re- duce this fluctuation (hallucination): Doesnt seem to be exclusive to pure debate the case from figure4 d-f

14. 4.1 conclusion 2: if that is true why is hallucination lower for P0P1P1 or P0P1P0

15. section 4.2 number of agents: doesnt seem to be always case on both accounts

16. section 4.2 more rounds: Doesn’t seem to be always the case. The pattern seems inconsistent as is defintion of good or bad. Also, the improvement needs statisitcal significance.

17. Section 4.2 Other Collaborative Strategies: Doesn’t seem to be always the case. All except for 1 case the performance consistently drops. Also, what is significant drop needs defintion (at least for one case where they are pretty close). And same with what is good or bad performance to begin with.

---

> ### Author Response · Authors · 2023-11-18
> **Response to Reviewer pe1t (Part 1)**
>
> Dear Reviewer pe1t,
>
> Thank you for your insightful feedback and constructive questions regarding our manuscript. We appreciate the opportunity to clarify and elaborate on the points you raised. Below, we address each of your questions (the same as weakness points you proposed) individually.
>
>
> ## Response to Q1:
>
> The composition of agents with different 'traits' (easy-going or overconfident) in the society, even without marked differences, provides a nuanced understanding of social dynamics, with Sherif’s Classic Autokinetic Effect Study[1] being a representative piece of research. It serves to model real-world scenarios where subtle variations in behavior or capability can significantly impact collective outcomes. For example, as can be seen from Table 2, the performance exhibited by different societies is not always exactly consistent and still present differences.
> Our design of society’s agent composition initially did not account for the lack of significant differences among different societies. However, this phenomenon became apparent after our simulation experiments.
> Besides, Reviewer wLih thinks the assertion that collaborative strategies overshadow the influence of agent composition may be premature.
>
> Moreover, our supplementary experiments in the Appendices (as illustrated in Figure 8, where we conducted a word frequency analysis on two societies with the most distinct compositions, S1 and S4) reveal that all societies tend to harmonize. This suggests a propensity of agents to exhibit easy-going rather than overconfident traits.
> We speculate that this could be due to issues with the prompt or the model (similar to what you mentioned in Q4), involving two potential scenarios:
> - (1) The model itself is not at fault; rather, the ineffectiveness of the prompt is the issue.
> - (2) The prompt is adequate, but the model does not adhere to it.
> We lean towards the scenario (2) as the more probable cause, and you can check our response to Q4 for more details, where we found that the prompt is effective enough.
>
> Furthermore, we relate this phenomenon to the alignment of LLMs, which discourages the adoption of traits like overconfidence that conflict with human preferences. This indicates a need for LLMs to strike a balance between "following instructions" and "maintaining non-toxicity", instead of only trying to mirror human preferences.
>
>
> ## Response to Q2:
>
> "on MMLU S1" you mentioned is just one case and cannot adapt to all of the scenarios. To ensure the differences in performance are statistically significant and not by chance, we conduct additional ANOVA analysis, to provide a more rigorous significance test for these comparisons in the main experiment of section 3.1. These findings have been included in Table 4, Appendix F.1 of the revised manuscript.
>
> A notable observation is that the p-value associated with the "collaborative strategy" is significantly below the 0.05 threshold, indicating its substantial impact. In contrast, the p-value of the other two factors, "society" and "collaborative strategy and society", is obviously greater than 0.05. This corroborates our earlier conclusion in section 3.1, emphasizing that **the influence of collaborative strategy outweighs that of society, and performance and collaborative strategy are significantly correlated.**
>
>
> ## Response to Q3:
>
> Based on the results in Table 2, it's evident that the collaborative strategy $p_0p_0p_1$ is not always superior to $p_0p_0p_0$, but there are some collaborative strategies that achieve performance close to $p_0p_0p_0$. On MATH S3, $p_0p_0p_1$ actually performs worse. Only for MMLU dataset, $p_0p_0p_1$ tends to be the optimal choice, as shown in Table 2 and Figure 4(d). We observe that in comparison to $p_0p_0p_0$, $p_0p_0p_1$ increases the proportion of `correcting mistakes` while maintaining the rate of `changing correct answers`. Additionally, Figure 12 illustrates that $p_0p_0p_1$ achieves a consistency of 2.8 in the first two rounds. At this point, the magnitude of changes in answers triggered by one reflection ($p_1$) is relatively minor (as shown in Figure 4(d)). This scenario can be akin to trials in human society (assuming the stability of human society), where new trials invariably lead to fresh discoveries and innovations, thereby advancing human science and technology.
> The analysis in our response to Q7 also helps to understand this issue.
>
> \\
>
> References
>
> [1] https://opentext.wsu.edu/social-psychology/chapter/module-7-social-influence/

---

> ### Author Response · Authors · 2023-11-18
> **Response to Reviewer pe1t (Part 2)**
>
> ## Response to Q4:
>
> We enhanced our experiments to validate the effectiveness of the Prompt and Model, maintaining alignment with the main experiments detailed in this paper.
> Detailed experimental setup is available in Appendix C: for the prompt, refer to Table 3 in the Appendix; and for complete dialogue examples, refer to Figures 9 and 10 in the Appendix. The experiments used the gpt-3.5-turbo model, with both overconfident and easy-going agents using 0-shot prompting.
>
> **Effectiveness of the Prompt**: Analysis of the word cloud in Figure 8 suggests that the prompt designed for easy-going traits is successful. Consequently, our attention turns to confirming the effectiveness of the prompt for overconfident traits. In the absence of a robust validation approach, we draw upon our existing experimental results and practical experience to assess the prompt's effectiveness from four key angles:
>
> - *Behavior Explanation*: We identified two behaviors, "confident in your answer" and "persuades other agents to believe in you", which align with the behavioral display of overconfidence.
> - *Refusal to Accept*: We employ a role-play method to prompt the model, and then ask if the model comprehends the prompt (if a prompt induces harmful content output, the model will refuse it). When checking our logs, the model didn't refuse our prompts and replied affirmatively with 'OK'.
> - *Re-asking*: Building on the previous step, we posed another query: "What should you do if someone else's answer differs from yours?". The model's reply, "I will convince them", proves the effectiveness of our prompt tailored for overconfident traits.
> - *Example Analysis*: Delving deeper, we contextualize the "re-asking" with a practical scenario. Even if the model's own response is wrong and our prompted answer is correct, the model still insists on its viewpoint. This observation further substantiates the effectiveness of our prompt tailored for overconfident traits.
>
> **In summary, these experiments collectively affirm the effectiveness of the prompts we have developed for both easy-going and overconfident traits.**
>
>
> ## Response to Q5:
>
> (1) For *details regarding the experimental setup and task specifics*, please refer to our response to Q4, with extensive information located in Appendix C.
>
> (2) Regarding *the number of prompts required for the agent to reach an answer during reflection*, there is no definitive count. The nature of reflection, devoid of direct feedback and influenced by human biases, results in the agent's answers evolving throughout the reflection process.
>
> (3) On the matter of *whether the test data was exposed during the training phase*, this topic has been extensively debated. For further insights, refer to [2]. The training of GPT3.5 model involves three distinct phases: pre-training, Supervised Fine-Tuning (SFT), and Alignment, detailed at https://openai.com/blog/chatgpt. It is possible that certain stages of training could lead to some information being forgotten. In a hypothetical extreme scenario where test data is present at all three stages, the likelihood of answer variation for a single-turn dialogue in response to a query is low, typically yielding a high accuracy rate. However, by introducing a reflection prompt, which creates a multi-turn dialogue, the experimental findings indicate that the reflection process induces notable alterations in the responses.
>
> (4) The *main hypothesized causes for these changes during reflection* are twofold: (i) the absence of constructive feedback during self-reflection, and (ii) a propensity for altering initial responses during the alignment phase when prompted to reflect. As illustrated in Figure 12, there is a marked reduction in answer consistency on the Chess dataset following reflection. This is further evidenced by the prevalent use of the word 'apology' in the word cloud depicted in Figure 8. In essence, when prompted to reflect, the model tends to reassess its own responses for potential errors, a tendency that becomes more evident when the model encounters unfamiliar question types.
>
>
> ## Response to Q6:
>
> Sure, the three datasets exhibit substantial differences. MMLU is characterized by knowledge-based, discrimination, and straightforward questions. The MATH dataset comprises reasoning, generative, and challenging problems. Meanwhile, Chess dataset features spatial imagination, generative, and complex problems, including those with multiple potential answers. As for *the differences across the three datasets*, we compare MATH vs MMLU and Chess move validity, not one vs another one between all three tasks, due to space limits.
>
>
> \\
>
> References
>
> [2] Machine Psychology: Investigating Emergent Capabilities and Behavior in Large Language Models Using Psychological Methods.

---

> ### Author Response · Authors · 2023-11-18
> **Response to Reviewer pe1t (Part 3)**
>
> ## Response to Q7:
>
> (1) The behavior described in section 3.2 does indeed present a contradiction, mirroring a similar phenomenon in human society.
>
> (2) In terms of *collaboration times*, as elaborated in section 3.2 and illustrated in Figures 4(a-c), multiple collaboration proves to be counterproductive for MMLU tasks, as accuracy drops after 3-round collaboration. However, for more complex tasks, such as MATH, 3-round collaboration can be advantageous. We also conducted additional experiments to thoroughly analyze the evolution of conformity phenomenon across varying collaboration times, and this analysis is detailed in Figure 11 of the Appendix. For this analysis, we focused specifically on the turn of debate ($p_0$), where conformity is more prevalent, and compiled statistics based on individual participants.
> Firstly, a broad view reveals that the prevalence of conformity is over 50%, suggesting its widespread nature. Secondly, we classified the shifts in the correctness of answers, pre- and post-conformity, into four categories: True->False, False->True, True->True, and False->False. The results from Figure 11 reveals that the proportion of correct answers post-conformity outweighs that of incorrect answers, indicating the potential benefits of conformity. However, an interesting trend emerges with an increasing number of turns/rounds: the proportion of True->False answers rises, while the False->True ratio remains constant. Moreover, the increment in True->True answers is less than that in False->False, suggesting a diminishing return with an excess of interaction turns/rounds.
>
> (3) For *factual information can be recalled/learn from training data vs the problem where answer may not be factual but available as a set of rules*, we guess you mean how to achieve the final answers if there are multiple candidate answers that may contain factual information and rules of induction. An explanation for this can be found in Appendix C.2. Taking the Chess dataset as an example, which permits multiple correct answers, we initially evaluate the accuracy of each individual’s response. Subsequently, a majority vote based on these accuracy evaluations is conducted. For instance, if the correct solutions to a problem are [e1, e3], and the answers provided by three agents at a certain moment are [e1, e3, b1], we classify these as [Correct, Correct, Wrong], and then decide the overall correctness of the discussion. In the case of the MATH dataset, assume the correct answer is 1/4; however, answers such as 2/8 or 0.25 are also deemed correct. This judgment is reached through a combination of manual review and rule-based matching.
>
>
> ## Response to Q8:
>
> We utilized a zero-shot setting for our experiment. To determine the answer for a specific round, we relied on a majority vote, taking into account the responses from all agents within a society for that round. This approach essentially harnesses the collective wisdom of LLMs to yield better answers.
>
> Moreover, there are various methods to decide on an answer. It could either be determined by one of the participants or an external observer. We opted for the latter, choosing majority vote for two key reasons: firstly, this method is widely used in numerous studies [3]; secondly, graph theory [4] suggests that when over two-thirds of the participants concur on an answer, it often leads to a more optimal result, a concept also prevalent in most electoral systems.
>
> Your idea of tailoring the answer determination method to the specific composition of a society is intriguing. Our present experiments indicate no substantial differences between different societies, thereby justifying the use of majority vote. Should notable variances between societies arise in the future, we might consider adapting our decision-making strategy to reflect these societal compositions. For example, increasing the weight of answers from overconfident agents could be one such adaptation.
>
> \\
>
> References
>
> [3] Improving Factuality and Reasoning in Language Models through Multiagent Debate.
>
> [4] Practical Byzantine Fault Tolerance. OSDI.

---

> ### Author Response · Authors · 2023-11-18
> **Response to Reviewer pe1t (Part 4)**
>
> ## Response to Q9:
>
> - *"How can we be sure this is from hallucination?"*: In Section 4, we define TFTF, FTFT, TFFT, and FTTF as "Wavering Answers", as they represent answers that continually oscillate (taking MMLU as an example, the answer may be A at one moment, then B at another, and possibly C later). This kind of conflict is considered a manifestation of model hallucination [5].
> - *"Not model changing it’s output to what user prefers from set of possibilities?"*: Our tasks don’t fall under creative endeavors such as novel writing. Rather, it is characterized by definitive answers with a limited set of possible outcomes. In our evaluation process, the emphasis is placed on the accuracy of the answers, not on the composition of the sentences. Given that the answers are objective in nature, they do not accommodate subjective human preferences.
>
>
> ## Response to Q10 and Q11:
>
> We conducted a two-way ANOVA analysis of significance test (before that we ensure that it conformed to normal distribution and variance homogeneity), and the results are shown in Table 4 at Appendix. We found that only the collaborative strategy had a significant impact, while the influence of society was not significant.
>
>
> ## Response to Q12:
>
> It seems there was a misunderstanding due to previous statements. The conclusion refers to the observation that, for 'answers wavering', the proportion of continuous reflection $p_1$ (which implies more severe model hallucination) is the highest.
> - On MMLU dataset, the top three proportions are $p_1p_1p_1$、$p_1p_0p_1$、$p_1p_1p_0$;
> - For MATH dataset, the top three are $p_1p_1p_1$, $p_1p_1p_0$, $p_0p_1p_1$;
> - On Chess dataset, the leading three are $p1p_0p_1$, $p_1p_1p_1$, $p_0p_1p_1$.
>
>
> ## Response to Q13:
>
> We've made some clarifications to address potential ambiguities in the original text. Debates are shown to effectively reduce model hallucinations. When examining the strategies with the lowest rates of wavering answers, on MMLU, MATH, and Chess datasets, the strategies with the lowest are respectively $p_0p_0p_0$, $p_1p_0p_0$, $p_0p_0p_0$. Notably, for the MATH dataset, the difference between $p_0p_0p_0$ and $p_1p_0p_0$ is minimal. Furthermore, in terms of consistency, as depicted in Figure 12 at Appendix, on all three datasets, the consistencies of $p_0p_0p_0$ and $p_1p_1p_1$ represent two contrasting extremes.
>
> \\
>
> References
>
> [5] HaluEval: A Large-Scale Hallucination Evaluation Benchmark for Large Language Models.

---

> ### Author Response · Authors · 2023-11-18
> **Response to Reviewer pe1t (Part 5)**
>
> ## Response to Q14:
>
> In the revised version, we have changed the original description to "debates can effectively reduce hallucinations."
> As for "hallucination lower for $P_0P_1P_1$ or $P_0P_1P_0$", we guess you mean that based on Figure 4(d-f), it is observed that for different collaborative strategies, $p_0p_1p_1$ and $p_0p_1p_0$ tend to have lower hallucinations on MMLU dataset. This conclusion should be evaluated comparatively. In the case of these two strategies (both starting with $p_0p_1$ interactions in the first two rounds), they align with the observed trend.
> - The $p_0p_1p_0$ strategy, not involving continuous reflection, exhibits a lesser proportion of hallucinations compared to $p_0p_1p_1$.
> - Conversely, the $p_0p_1p_1$ strategy, which includes continuous reflection, shows a higher proportion of hallucinations compared to $p_0p_1p_0$. This pattern is consistently observed across all three datasets.
>
>
> ## Response to Q15:
>
> Regarding *the Number of Agents*, we conducted a one-way ANOVA analysis of significance, as shown in Table 5 at Appendix. We found that the increase in the number of agents does not present a significant impact.
>
>
> ## Response to Q16:
>
> Regarding "Different Rounds," we employed a one-way ANOVA analysis for testing, as detailed in Table 6 at Appendix. From the combined insights of Figure 6 and Figure 12, we noted that strategies with lower consistency and greater variability are more sensitive to the number of rounds. This is especially true for the continuous reflection strategy on Chess dataset, where continuous reflection results in a gradual decrease in consistency, meaning lesser fluctuations and more uniformity, thus showing less sensitivity to the number of rounds. Strategies with the p-value greater than 0.05 generally exhibit higher consistency. However, increasing the number of rounds does not always equate to improved outcomes. For strategies that already have high consistency, adding more rounds can actually decrease performance, as illustrated in Figure 11 in Appendix. This is also discussed in our response to Q7.
>
>
> ## Response to Q17:
>
> Regarding "Other Collaborative Strategies," we applied a one-way ANOVA analysis, as indicated in Table 7 at Appendix. In this context, 'performance' pertains to accuracy, with higher accuracy denoting better performance. Our findings reveal that collaborative strategies significantly affect Chess dataset. In contrast, their impact on the MMLU dataset is less pronounced. This is primarily because collaboration doesn't substantially benefit the MMLU dataset, as shown in Figure 4(a), and as further elaborated in our response to Q7.
>
> \\
>
> References
>
> [1] https://opentext.wsu.edu/social-psychology/chapter/module-7-social-influence/
>
> [2] Machine Psychology: Investigating Emergent Capabilities and Behavior in Large Language Models Using Psychological Methods.
>
> [3] Improving Factuality and Reasoning in Language Models through Multiagent Debate.
>
> [4] Practical Byzantine Fault Tolerance. OSDI.
>
> [5] HaluEval: A Large-Scale Hallucination Evaluation Benchmark for Large Language Models.

---

> > ### Author Response · Authors · 2023-11-22
> > **Gentle Reminder and Appreciation for Continued Participation in Manuscript Review Discussion to Reviewer pe1t**
> >
> > Dear Reviewer pe1t,
> >
> > We would like to express our profound gratitude for your insightful review. Following your valuable suggestions, we have replied to your questions (the same as weaknesses as you proposed) and updated our submission accordingly.
> >
> > Could we kindly enquire if our responses and adjustments have adequately resolved your concerns?
> > We are more than happy to answer any further queries or concerns you may have. Thank you once again.
> >
> >
> > Best Regards,
> >
> > Authors

---

### Author Response · Authors · 2023-11-18
**Official Response to All Reviewers**

Dear Reviewers,

We sincerely appreciate the time and effort you have dedicated to reviewing our manuscript. Your insightful comments and suggestions are valuable in enhancing the quality and clarity of our research.

We have uploaded a revised version of our paper that incorporates the suggestions in the reviews. **The changes are marked in blue, and most of the revised content and additional experiments are at Appendix, especially in Appendix E and F.**

The revised content mainly contains
- Polished expressions of experimental analysis, especially in Section 4.
- Key Takeaways for Multi-Agent Collaboration
- Additional experiments: ANOVA analysis for significance test the main experiments
- More detailed analysis on Conformity and Consistency
- More detailed analysis on the Number of Agents, Rounds, Collaborative Strategies in society settings
- Exploration on Effectiveness of Prompt
- Potential Real-world Applications
We believe the new version is clearer based on the valuable suggestions. Please let us know if our responses address your questions or concerns. We will be very happy to respond to further questions. Thank you all once again.

*To view the response and the revision, please click here:* https://openreview.net/forum?id=ueqTjOcuLc.


Best Wishes,

Authors

---

### Author Response · Authors · 2023-11-22
**Gentle Reminder and Appreciation for Continued Participation in Manuscript Review Discussion**

Dear Reviewers and AC,

We genuinely value the constructive comments and insightful suggestions you provided for our work. Recognizing the approaching end of the discussion period on November 22nd, we kindly remind you to participate in the ongoing discussion and provide any additional insights or clarifications you may have. Your expertise is invaluable to us, and we believe your input will significantly contribute to the improvement of our work.

Thank you very much for your time and consideration. We look forward to hearing from you soon.


Best Wishes,

Authors

---

### Meta-Review · Area_Chair_3w7q · 2023-12-08

**Metareview:**

This paper looks at the collaboration mechanisms of LLM agents. By making "societies" of LLM-based agents, this paper attempts to uncover how they exhibit behavior similar to humans in collaboration, from a social psychology perspective. While the topic and approach are quite interesting, as the reviewers point out, this paper needs more work before publication. I appreciate the uploaded revision with additional experiments, but these revisions are very major and need another full round of review. I strongly encourage the authors to resubmit to a future venue.

**Justification For Why Not Higher Score:**

The authors respond to the reviewers' comments and questions, some with direct revision, but most with promise to revise in the final version. The revisions are major such that just saying these will be revised is not sufficient to raise the overall score. Accepting this paper would require additional experiments, better (more rigorous) analyses and interpretation of the results.

**Justification For Why Not Lower Score:**

N/A

---

### Decision · Program_Chairs · 2024-01-16

Reject